



# Variability in copepod trophic levels and in feeding selectivity based on stable isotope analysis in Gwangyang Bay off the southern coast of Korea

Mianrun Chen[1,2], Dongyoung Kim[2], Hongbin Liu[3], Chang-Keun Kang[1]

[1]South China Sea Institute of Planning and Environmental Research, South China Sea Branch, SOA, Guangzhou, P.R. China
[2]School of Earth Sciences & Environmental Engineering, Gwangju Institute of Science and Technology, Gwangju 61005, Republic of Korea
[3]Division of Life Science, The Hong Kong University of Science and Technology, Clear Water Bay, Kowloon, Hong Kong SAR

*Correspondence to*: Chang-Keun Kang (ckkang@gist.ac.kr)

**Abstract.** Trophism (i.e., food resources and trophic levels) of different copepod groups was assessed along a salinity gradient in the temperate estuarine Gwangyang Bay of Korea, based on seasonal investigation of taxonomic results in 2015 and stable isotope analysis incorporating multiple linear regression models. The $\delta^{13}$C and $\delta^{15}$N values of copepods in the bay displayed salinity-associated spatial heterogeneity as well as temperature-related seasonal variations. Both spatial and temporal variations reflected those in isotopic values of food sources. Three major groups (marine calanoids, brackish water calanoids and cyclopoids) had a mean trophic level of 2.2 relative to nanoplankton as the basal food source, similar to the bulk copepod assemblage; however, they had dissimilar food sources based on the different $\delta^{13}$C values. Calanoid isotopic values indicated a mixture of different genera including species with high $\delta^{15}$N values (e.g., *Sinocalanus* and *Labidocera*) and relatively low $\delta^{15}$N values (*Paracalanus* and *Acartia*). Feeding preferences of different copepods probably explain these seasonal and spatial patterns of the community trophic niche. Bayesian mixing model calculations based on source materials of two size fractions of particulate organic matter (nanoplankton at <20 m vs microplankton at 20–200 m) indicated that *Acartia* preferred large particles, *Paracalanus* and *Pseudodiaptomus* apparently preferred small particles, and Corycaeus was typically omnivorous with low selectivity on particle size. In addition, the carnivorous genus *Tortanus* predated on copepods without apparent selectivity, *Labidocera* preferred *Acartia* to *Paracalanus*, and *Sinocalanus* preferred *Paracalanus* to *Acartia* and cyclopoids. Overall, our results depict a simple energy flow of the planktonic food web of Gwangyang Bay: from primary producers (nanoplankton) and a mixture of primary producers and herbivores (microplankton), through omnivores (*Acartia*, *Paracalanus*, and *Corycaeus*) and detrivores (*Pseudodiaptomus* and *Euterpina*) to carnivores (*Tortanus*, *Labidocera*, and *Sinocalanus*).

## 1 Introduction

Mesozooplankton constitute essential trophic mediators of marine food webs in transferring energy and materials by linking





the microbial food web to higher trophic levels. Copepods are a diverse assemblage dominating mesozooplankton communities. With broad feeding spectra and flexible feeding strategies, the bulk copepod assemblage is omnivorous, depending on dominant species or groups (Graeve et al., 1994; Sell et al., 2001; Turner, 2004; Vadstein et al., 2004; Gifford et al., 2007; Chen et al., 2017). The role of copepods in planktonic food webs can be determined by their overall trophic

levels (TLs) relative to primary producers. In turn, the TLs of a diverse copepod assemblage are balanced from different groups with different feeding preferences and are ultimately determined by species composition. Because copepods play a fundamental role in feeding on phytoplankton as primary consumers, so the seasonal and spatial changes in the composition and availability of phytoplankton determine the abundance and feeding behavior of the copepod assemblages.

The most dominant copepod species, such as *Neocalanus*, *Calanus*, *Temora*, and *Paracalanus*, are filter-feeders that perform

a size-selective feeding behavior depending on particles effectively retained by feeding appendages of copepods. Large phytoplankton (> 20 μm; mainly diatoms and dinoflagellates) are generally grazed at high rates by copepods, as shown by many field studies in coastal and estuarine waters (e.g., Liu et al., 2005a,b; Chen et al., 2017). Many other field studies have reported that omnivorous species dominate copepod assemblages because of high feeding selectivity on larger microzooplankton that are considered to have higher nutritional quality (e.g., Berk et al., 1977; Fessenden and Cowles, 1994;

Calbet and Saiz, 2005; Gifford et al., 2007; Chen et al., 2013). These omnivorous copepods might induce increases in phytoplankton levels indirectly through trophic cascades as they graze intensely on microzooplankton (e.g., ciliates and heterotrophic dinoflagellates) (Nejstgaard et al., 2001; Stibor et al., 2004; Sommer and Sommer, 2006; Zöllner et al., 2009; Chen et al., 2011, 2013). Therefore, the assessment of the trophic position (herbivores, omnivores, or carnivores) of copepods within a complex planktonic food web is critical in predicting the ecological relationships between predators and

prey.

Stable isotope analysis (SIA) is a reliable technique providing insight into the trophic positions of copepods relative to basal food sources (Grey et al., 2001; Sommer et al., 2005; Hannides et al., 2009; Kürten et al., 2011). Isotopic comparisons with food sources enable us to analyze prey selectivity during predators' feeding history as well as within food web structures (Fry, 2006; Layman et al., 2012). In general, the carbon stable isotope ratio ($\delta^{13}$C) can be useful for tracing food sources

because of small fractionation (0.5−1‰ per TL) during trophic transfer, particularly when different food sources at a given period in a specific system have distinct $\delta^{13}$C values. By contrast, the nitrogen stable isotope ratio ($\delta^{15}$N) can be useful for estimating relative TLs because $\delta^{15}$N values of consumers generally increase with TL (an average 3.2‰ of enrichment per TL; Post, 2002; Michener and Kaufman, 2007). The development of linear mixing models and the Bayesian mixing model has allowed researchers to predict the proportions of different food sources in the diets assimilated by grazers (Phillips and

Koch, 2002; Phillips and Gregg, 2003; Moore and Semmens, 2008; Ward et al., 2010; Parnell et al., 2010, 2013).

Coastal and estuarine environments often experience rapid fluctuations of inorganic carbon and nitrogen inputs in response to diverse oceanographic processes (e.g., coastal currents, upwelling, tidal mixing, and river discharges), which drive spatial and seasonal heterogeneities in biogeochemical dynamics and isotopic signatures (Rolff, 2000). Indeed, the $\delta^{13}$C values of




suspended particulate organic matter (POM) in estuarine systems increase progressively from the head to the mouth of each estuary because of the lower $\delta^{13}C$ values in terrestrial carbon or sewage materials through river discharge (Cifuentes et al., 1988). In contrast, the $\delta^{15}N$ values of primary producers increase from being nutrient-sufficient (high fractionation) to nutrient-limiting (low fractionation) and are especially high in anthropogenic wastewater nitrogen inputs (McClelland et al., 1997). In addition, different phytoplankton groups—including diatoms, flagellates, dinoflagellates, and cyanobacteria— utilize different nitrogen sources with different enrichment factors, possibly offering different isotopic pools to grazers (Gearing et al., 1984; Rolff, 2000; Montoya et al., 2002).

Given that isotopic values of copepods vary in association with copepods' food source by one or two increases in TL values, seasonal and spatial patterns generally follow the trends of their food sources or dominant prey (Grey et al., 2001; Montoya et al., 2002; Kürten et al., 2011). Higher $\delta^{15}N$ values of copepods caused by fractionation rather than food source or by averaging from mixed food sources are evident considering the lowered isotopic values of fecal pellets (Checkley and Entzeroth, 1985; Checkley and Miller, 1989; Tamelander et al., 2006). Furthermore, the effect of the microbial food web on the elevated $\delta^{15}N$ values of copepods cannot be ignored (Rolff, 2000; Kürten et al., 2011). Therefore, variations in isotope signatures of both copepods and POM (including phytoplankton, bacteria, ciliates, and detritus) help to depict the biogeochemical cycles of specific systems (Grey et al., 2001; Montoya et al., 2002; Francis et al., 2011). Nevertheless, because copepods graze preferentially on larger phytoplankton (diatoms and dinoflagellates) and microzooplankton (ciliates and heterotrophic dinoflagellates), we hypothesize that isotopic values of the copepod assemblage will be much closer to those of larger rather than smaller food source plankton.

However, highly mixed species and size overlap make it hard to determine the relative trophic positions of different subgroups or species. Isotope analysis for different subgroups requires great expertise in isolating species from highly complex mixtures. Moreover, the number of individuals of a specific genus is often insufficient for analysis because of limited instrument sensitivity. Thus, to our knowledge, direct comparisons of different mesozooplankton groups or copepod species are seldom found in the literature (Schmidt et al., 2003; Sommer et al., 2005; Hannides et al., 2009). Here, we estimated isotope values of different copepods by mass balancing linear mixing models from values of bulk samples and taxonomic data of copepods. The allocated masses of calanoids and cyclopoids were achieved from random body ratios computed using R software.

Overall, we aimed (1) to understand the seasonal variations and spatial heterogeneity of copepod $\delta^{13}C$ and $\delta^{15}N$ values in a temperate estuarine system (Gwangyang Bay, Korea); (2) to compare the trophic positions of different copepods; and finally (3) to elucidate the compositions of two major size classes (<20 μm and >20 μm) of POM in grazer diets. The dietary composition (nano- vs microplankton) of copepods was estimated using Bayesian isotopic mixing models (Parnell et al., 2010, 2013). The results of this study will provide insights into trophism information (i.e., food resources and trophic levels) for different copepod groups and help in understanding the biogeochemistry of this estuarine system




## 2 Materials and methods

### 2.1 Study area

Gwangyang Bay is a semi-enclosed bay system, located on the southern coast of the Korean Peninsula, and is one of the most industrialized coastal areas exposed to anthropogenic pressure. It starts from Seomjin River through Yeosu Channel

(between Yeosu Peninsula and Namhae Island) to open ocean (the East China Sea). The bay area covers approximately 145 km$^2$ and water depth is generally shallow at 2.4–8.0 m in the northern upper-middle Seomjin River estuarine channel compared with 10–30 m in the deep bay channel (Kim et al., 2014). The annual freshwater discharges of Seomjin River are 10.7–39.3 × 10$^8$ tons. The seasonality of nutrient input from the catchment area (ca. 5 × 10$^3$ km$^2$), including agricultural and forested land, is profound (Kwon et al., 2002). The wet season starts from late spring and the discharge peaks during the

summer monsoon period.

Accordingly, the maximum median river discharge varies from 30–95 m$^3$ s$^{-1}$ in the dry season to 300–400 m$^3$ s$^{-1}$ in the summer monsoon, with an annual mean of c. 120 m$^3$ s$^{-1}$ (Kim et al., 2014). The tidal cycle of the bay is semidiurnal with maximum ranges of 3.40 m during spring tides and 1.10 m during neap tides. Tidal currents from the Yeosu Channel also strongly influence the system and approximately 82% of the Seomjin River flux is discharged toward this channel. Overall,

increasingly industrial pollution facilitates eutrophic conditions in the estuarine and related bay waters. Diatoms dominate in the phytoplankton community and density is high in the middle of Gwangyang Bay (Kim et al., 2009; data from our parallel study not shown). The distribution patterns of copepods in the Seomjin River during summer were represented by three main salinity zones: an oligohaline zone (predominated by *Pseudodiaptomas koreanus*, *Sinocalanus tenellus*, and *Tortanus dextrilobatus*), a mesohaline zone (predominated by *Acartia ohtsukai* and *Acartia forticrusa*), and a polyhaline zone

(predominated by *Acartia erythraea, Calanus sinicus, Centropages dorsispinatus, Labidocera rotunda*, and *Paracalanus parvus*) (Park et al., 2015)

### 2.2 Sampling and processing

Surface water and net-tow samples were collected seasonally (February, May, August, and November) at nine stations from the head to the mouth of Gwangyang Bay in 2015 (Fig. 1). Stations 1–3 were located in the Seomjin River, stations 4 and 5

were in Gwangyang Bay (the middle part of the estuary), and stations 6–9 were located from the offshore deep-bay channel to the southern mouth of the estuary. On each sampling occasion, water temperature and salinity were determined in situ using a YSI Model 85 probe (YSI Inc., Yellow Springs, OH, USA).

Zooplankton taxonomic samples were collected by net towing using a plankton net (45 cm diameter, 200 μm mesh size) equipped with a flowmeter (Model 2030R Mechanical Flowmeter, General Oceanics Inc., Miami, FL) and gently hauled

horizontally at a subsurface depth of 0.5–1 m with the ship speed at about 1 knot (0.5 m s$^{-1}$). Samples were fixed in formalin solution with a final concentration of 5% and then identified and enumerated under a stereomicroscope (SMZ 645; Nikon, Tokyo, Japan) in the laboratory. At each station, one additional net tow was collected for isotope analysis. After collection,





specimens were transferred immediately into plastic bottles and preserved in a refrigerator (4 ℃) until analysis. In the laboratory, subsamples were picked out from the mixed zooplankton samples under a dissecting microscope. Easily distinguishable zooplankton groups such as harpacticoids and decapods were separated from a mixture of calanoids and cyclopoids. All subsamples were lyophilized and then homogenized by pulverizing them with a mortar and pestle before

isotope analysis.

POM in surface water (0.5–1 m depth) was collected using a 5-l Niskin bottle at a midday high tide at the same time as zooplankton collection. Approximately 20 l seawater collected was first screened through a 200 μm Nitex mesh to remove zooplankton and large-sized particles. The prescreened water samples were transported to the laboratory as soon as possible. In the laboratory, water samples were filtered again through a 20 μm Nitex mesh and then filtered onto precombusted

(450 ℃ for 4 h) Whatman GF/F glass fiber filters to determine isotope ratios of fine POM (< 20 μm) representing pico- and nano-sized phytoplankton. To obtain enough plankton cells for isotope analysis of coarse POM (≥ 20 μm), we collected POM samples by net towing with a plankton net of 50 cm diameter and 20 μm mesh size. After collection, each sample was prefiltered through a 200 μm Nitex mesh to remove large particles and zooplankton. Both size fractions of samples were prepared in duplicate. Samples for $\delta^{13}C$ measurements were acidified by fuming for about 5 h over concentrated HCl in a

vacuum desiccator to remove carbonates, while the samples for $\delta^{15}N$ measurements were not acidified. All the samples were lyophilized and pulverized with a mortar and pestle before isotope analysis.

For chlorophyll *a* (Chl *a*) determination, 1-l subsamples of surface water were filtered through Waterman GF/F glass fiber filters. The filters for Chl *a* (including other photosynthetic pigments) were extracted with 95% methanol (5 ml) for 12 h in the dark at −20 ℃ and sonicated for 5 min to foster cell disruption. Aliquots of 1 ml of the supernatants were mixed with

300 μl of water; 100 μl of this solution was analyzed by reverse-phase high performance liquid chromatography (HPLC, LC-20A HPLC system, Shimadzu Co., Kyoto, Japan) using a Water Symmetry $C_8$ (4.6 × 150 mm, particle size: 3.5 μm, 100 Å pore size) column (Waters, Milford, MA, USA) and a method derived from Zapata et al. (2000). Quantification of standard pigments was calculated by spectrophotometer with the known specific extinction coefficients after Jeffrey at al. (1997). Sample peaks were identified based on their retention time compared with those of pure standards. Further details on

analysis, calibration, and quantification have been given elsewhere (Lee et al., 2011; Kwak et al., 2017).

**2.3 Isotope analysis**

For measurements of carbon and nitrogen stable isotope ratios, all pretreated samples were analyzed using a continuous-flow isotope ratio mass spectrometer (CF-IRMS; Isoprime100, Cheadle, UK) connected to an elemental analyzer (vario Micro cube, Hanau, Germany) following the procedure described by Park et al. (2016). Briefly, powdered samples were sealed in

tin combustion cups and filter samples were wrapped with a tin plate. All prepared samples were put into the elemental analyzer to oxidize at high temperature (1030 ℃). $CO_2$ and $N_2$ gases were introduced into the CF-IRMS with the carrier being helium gas. Data of isotope values are shown in terms of δX, indicating the relative differences between isotope ratios



of the sample and conventional standard reference materials (Vienna Pee Dee Belemnite for carbon, and atmospheric $N_2$ for nitrogen), which were calculated by the following equations: $\delta X = [(R_{sample}/R_{standard}) - 1] \times 10^3$, where X is $^{13}C$ or $^{15}N$ and $R_{sample}$ and $R_{standard}$ are the ratios of heavy to light isotope for samples and standards, respectively. International standards of sucrose (ANU $C_{12}H_{22}O_{11}$; National Institute of Standards and Technology (NIST), Gaithersburg, MD, USA) for

carbon, and ammonium sulfate ([$NH_4$]$_2SO_4$; NIST) for nitrogen, were used for calibration after analyzing every 5–10 samples. The analytical precision for 20 replicates of urea were approximately $\leq 0.2$‰ and $\leq 0.3$‰ for $\delta^{13}C$ and $\delta^{15}N$, respectively.

**2.4 Data analysis**

Because we did not measure the weights of calanoids and cyclopoids separately, we calculated them using simple models

based on the percentage of abundance assuming that the proportion we picked to do isotope analysis was the same as the proportion in samples for composition analysis. Assuming that the individual mass of calanoids and cyclopoids were proportionally different, we used the following equations to assign the weight for each group:

$$W_1 = \frac{P_1}{(P_1 + X \times P_2)} \times W, \tag{1}$$

$$W_2 = \frac{X \times P_2}{(P_1 + X \times P_2)} \times W, \tag{2}$$

where $W$, $W_1$ and $W_2$ indicate the weights of bulk samples, calanoids, and cyclopoids, respectively. $P_1$ and $P_2$ indicate the percentage of calanoids and cyclopoids in bulk samples, respectively. $X$ indicates the difference of individual mass of cyclopoids to calanoids. If $X = 1$, then the weight is directly assigned to the two groups by abundance percentage. Because cyclopoids are generally smaller than calanoids, we can predict that $X$ falls into a range of ($0.1 \leq X \leq 1$). We manually generated a vector of 250 random values for $X$ in the given ($0.1 \leq X \leq 1$) range using R software and then the mean $W_1$ and

$W_2$ of each sample were estimated. Among calanoids, the body mass of different genera was considered the same so that we could use the percentage of abundance to allocate the total weights of calanoids to different genera. Although it caused some underestimation for the larger copepods, they were not dominant in the samples.

Once the weights of different groups were assigned, we computed the isotope ratio for each group by multiple linear regression models as in the following equations:

$$m \times \delta^{13}C = m_1 \times \delta^{13}C_1 + m_2 \times \delta^{13}C_2 + m_3 \times \delta^{13}C_3 + m_X \times \delta^{13}C_X + error, \tag{3}$$

$$m \times \delta^{15}N = m_1 \times \delta^{15}N_1 + m_2 \times \delta^{15}N_2 + m_3 \times \delta^{15}N_3 + m_X \times \delta^{15}N_X + error, \tag{4}$$

where $m$ is the weight of the total community and $m_1 - m_X$ are the weight of different groups or genera of each group. $\delta^{13}C_1 - \delta^{13}C_X$ and $\delta^{15}N_1 - \delta^{15}N_X$ are the $\delta^{13}C$ and $\delta^{15}N$ values of each group or genus, respectively. We used R software to do the estimation using the whole sampling data set. Mean values and standard deviations are illustrated in figures. Sparse



genera such as *Euchaeta* and *Clausocalanus* were not used in the estimation, while insignificant results for *Calanus, Centropages, Oithona*, and *Oncaea* are not shown in figures because of the particularly high standard deviations of mean estimates.

Given that the isotopic values of consumers come from their diets and thereby from mixed proportions of different sources, the proportions of each source could be simulated by linear mixing models with a fractionation factor (also called a trophic enrichment factor). For instance, a mass balance mixing model is given by:

$$\delta^{13}C_{consumer} = f_1\delta^{13}C_{source1} + f_2\delta^{13}C_{source2} + \cdots + f_n\delta^{13}C_{source\,n} + \alpha_{Carbon}, \tag{5}$$

$$\delta^{15}N_{consumer} = f_1\delta^{15}N_{source1} + f_2\delta^{15}N_{source2} + \cdots + f_n\delta^{15}N_{source\,n} + \alpha_{Nitrogen}, \tag{6}$$

$$f_1 + f_2 + \cdots + f_n = 1, \tag{7}$$

where $f_1-f_n$ are the proportion of different sources, and $\alpha_{Carbon}$ and $\alpha_{Nitrogen}$ are trophic enrichment factors for $\delta^{13}$C and $\delta^{15}$N values, respectively.

Here, a Bayesian isotopic mixing model (available as an open source Stable Isotope Analysis package in R: SIAR) was performed to estimate the relative contribution of nanoplankton (defined by fine POM in the present study) and microplankton (coarse POM) to the copepod diets, as well as copepods to the carnivore diets (Parnell et al., 2010, 2013). The model assumes that each isotopic ratio of consumers follows the pattern of a Gaussian distribution with an unknown mean and standard deviation. The structure of mean values of consumers is a weighted combination of the food sources' isotopic values. The weights make up dietary proportions (given by a Dirichlet prior distribution). The standard deviation is divided up between the uncertainty around the fractionation corrections and the natural variability between all individuals within a defined group (Parnell et al., 2010, 2013). Because the values of consumers calculated from bulk copepod samples using the previous multiple linear regression models were only means and standard errors, we generated a vector consisting of 250 numbers for each group by a random normal distribution function. We then used the default iteration numbers (iterations = 500,000, burin = 50,000) provided by the SIAR package to perform our analysis. Fractionation factors used in the model estimation were calculated from TLs (with 3‰ for $\delta^{15}$N and 0.5‰ for $\delta^{13}$C per TL). TLs were calculated from the $\delta^{15}$N difference between consumer and source as follows (Post, 2002): ( $TL = 1 + (\delta^{15}N_{consumer} - \delta^{15}N_{source})/3$ ). Concentrations of isotope per mass among different diets (nanoplankton, microplankton, and major copepod genera) were not considered in this study. Model fitting was done via a Markov Chain Monte Carlo (MCMC) protocol that produces simulations of plausible values of the dietary proportions of each source. More details on model simulation can be found elsewhere (Parnell et al., 2010, 2013).

All statistical analyses were performed using R 3.4.0 software (https://cran.r-project.org/bin/windows/base/). Regression analyses of copepod isotopic values were performed by generalized additive models (GAMs) using the *mgcv* library (Wood and Wood, 2015). Data were smoothed by cubic regression splines and fitted by the family of Gaussian. One-way analysis of variance (ANOVA) was adopted to test seasonal differences in environmental factors and copepod abundances, and



Student's *t*-tests were used to test for significant differences in mean $\delta^{13}C$ and $\delta^{15}N$ values between nano- and microplankton. Before applying ANOVA and *t*-tests, the data were tested for normality of distribution and equal variance; significance was assumed at $P = 0.05$.

## 3 Results

### 3.1 Environmental variability and zooplankton abundances

Environmental factors including temperature, salinity, Chl *a* levels, copepod abundance, dominant species, and percentages of total copepods are shown in Table 1. Water temperature was significantly higher in summer and lower in winter (ANOVA, $P < 0.001$). Spatial variability of salinity was significant, with extremely low values at stations 1 and 2 (the river mouth) and then the values gradually increased to station 5 (the middle of the bay). Chl *a* concentrations were significantly higher in spring and summer than in winter and autumn (ANOVA, $P < 0.01$). Despite insignificant spatial variability, higher Chl *a* concentrations generally occurred in the middle of the bay. Seasonal variability of copepod abundance was significant (ANOVA, $P < 0.01$), with higher abundances in winter when temperatures and Chl *a* concentrations were low.

Seasonal and spatial variations of dominant species of copepods were apparent. The marine calanoid *Acartia* dominated at the river mouth to the middle part of the bay, while *Paracalanus* dominated at the mouth of the bay during winter. *Acartia* also dominated at the most highly saline stations in summer, except for station 7, where the community was dominated by *Labidocera rotunda*. A brackish water-preferring calanoid species, *Pseudodiaptomus,* dominated stations 1 and 2 at the river mouth in spring and another brackish calanoid species, *Sinocalanus*, dominated station 1 in autumn. At the river-mouth stations in summer, copepods were unexpectedly dominated by the marine calanoid species *Tortanus dextrilobatus*. The cyclopoid species *Corycaeus affinis* mainly dominated the most highly saline stations in spring and autumn.

### 3.2 Variability of plankton $\delta^{13}C$ and $\delta^{15}N$ values

The $\delta^{13}C$ values of size-fractionated plankton (< 20 and 20–200 μm) and mixed copepod samples showed distinct spatial variations in each season (Fig. 2A–C). The $\delta^{13}C$ values of nanoplankton (< 20 μm POM) ranged from −27.6 to −19.4‰ with a mean of −22.7‰ (Fig. 2A). For nanoplankton, the lowest $\delta^{13}C$ value was found at station 1 (the upper stream station of Seomjin River) in spring and the highest at station 9 (the mouth of the estuary) in summer. The $\delta^{13}C$ values of microplankton (20–200 μm POM) ranged from −26.3 to −17.8‰ with a mean of −20.8‰ (Fig. 2B), being significantly higher than those of nanoplankton (paired *t*-test, $t = 7.6$, $P < 0.001$). Its lowest $\delta^{13}C$ value was found at station 2 in spring and the highest at station 8 in winter. Overall, similar to nanoplankton, the microplankton $\delta^{13}C$ values were more negative at the river portion (stations 1–3) and less negative at the mouth of the estuary (stations 7–9). The $\delta^{13}C$ values of mixed copepods ranged from −24.9 to −16.4‰, with the lowest at station 1 in autumn and the highest at station 8 in summer (Fig. 2C). The spatial variability of copepod $\delta^{13}C$ values followed the pattern of POM $\delta^{13}C$ values. However, the copepod $\delta^{13}C$ values were





significantly higher than those of nanoplankton (paired $t$-test, $t = 8.6$, $P < 0.001$) and microplankton ($t = 3.1$, $P = 0.004$). Their $\delta^{13}C$ values were higher in summer and winter than in spring and autumn. At stations 1–3, river input lowered the $\delta^{13}C$ values of nanoplankton during the wet season (spring to summer). At stations 4–9, significantly lower $\delta^{13}C$ values were observed in autumn than in other seasons (ANOVA, $F = 13.4$, $P < 0.001$). For copepods, the autumn values were

5 significantly lower than those in other seasons (ANOVA, $F = 5.9$, $P = 0.004$). Overall, seasonal successions of winter–spring, spring–summer, and summer–autumn were apparent for all plankton groups.

The $\delta^{15}N$ values exhibited wider fluctuations than $\delta^{13}C$ values (coefficients of variation = 29.3% vs 9.0%, 21.5% vs 11.8%, and 18.8% vs 13.1% for nanoplankton, microplankton, and copepods, respectively). The $\delta^{15}N$ values of nanoplankton ranged from 3.2‰ (station 4 in summer) to 8.8‰ (station 1 in winter) with a mean of 5.6‰ (Fig. 2D). There were distinct patterns

in the three locations of the bay. The $\delta^{15}N$ values tended to decline with distance from the river mouth, then increased in the middle of the bay, and decreased again toward the mouth of the estuary. The nanoplankton $\delta^{15}N$ values were higher in winter than in other seasons (paired $t$-test, $t = 5.4$, $P = 0.001$ for spring; $t = 3.0$, $P = 0.017$ for summer; $t = 4.1$, $P = 0.004$ for autumn). A mean $\delta^{15}N$ value of microplankton (7.6‰), ranging from 4.8‰ (station 2 in spring) to 16.2‰ (station 6 in spring), was significantly higher than that of nanoplankton (paired $t$-test, $t = 4.9$, $P < 0.001$). The microplankton $\delta^{15}N$ values

were higher in summer than in other seasons (ANOVA, $F = 4.6$, $P = 0.009$), with the spatial trend vanishing in summer. Indeed, spatial trends differed between seasons, increasing progressively from the river mouth to the bay mouth in spring and autumn, and decreasing in winter. Copepod $\delta^{15}N$ values ranged from 6.6 to 12.3‰ and were higher in summer than in other seasons (ANOVA, $F = 15.6$, $P < 0.001$), being much more consistent with the pattern of microplankton than that of nanoplankton.

Generalized additive model analysis showed that the deviances of copepod $\delta^{13}C$ and $\delta^{15}N$ values explained by the GAMs were 92.7% and 76.9%, respectively. Copepod $\delta^{13}C$ values changed significantly toward increasing salinity (GAM, $F = 7.9$, $P = 0.005$), for both the nanoplankton (GAM, $F = 6.2$, $P = 0.008$) and the microplankton (GAM, $F = 16.4$, $P < 0.001$; Fig. 3A–C). In contrast, temperature was the most important factor to explain the variability of copepod $\delta^{15}N$ values (GAM, $F = 13.6$, $P < 0.001$; Fig. 3D). The microplankton $\delta^{15}N$ value was another important contributor to the variability of copepod

$\delta^{15}N$ values (GAM, $F = 3.5$, $P = 0.034$; Fig. 3E), while nanoplankton $\delta^{15}N$ was not (GAM, $P > 0.05$). The Chl $a$ concentration influenced the variability of copepod $\delta^{15}N$ values significantly (GAM, $F = 3.3$, $P = 0.047$; Fig. 3F).

### 3.3 Trophic positions of major groups

Multiple linear regression analyses to estimate mean isotopic values of different copepod groups (i.e., brackish water calanoids, marine calanoids, and cyclopoids) from mixed copepod values (excluding harpacticoids) were all significant ($R^2 =$

0.94, $P < 0.001$ for $\delta^{13}C$; $R^2 = 0.78$, $P < 0.001$ for $\delta^{15}N$). The values of harpacticoids and decapods isolated from the winter and spring samples (stations 3, 4, 6, and 7) were measured directly. The three major groups displayed a similar mean $\delta^{15}N$ value (9.2‰) but contrastingly different $\delta^{13}C$ values ($-26.1 \pm 1.8‰$, $-20.6 \pm 1.6‰$, and $-22.1 \pm 1.8‰$ for brackish water




calanoids, marine calanoids, and cyclopoids, respectively; Fig. 4A). Their $\delta^{15}$N values were higher than those of food resources (nanoplankton and microplankton). The mean $\delta^{15}$N values of decapods (7.1 ± 1.5‰) and harpacticoids (6.9 ± 0.6‰) were lower than those of other copepods and microplankton. However, it should be noted that the mean $\delta^{15}$N value (8.0 ± 0.8‰) of a mixture of three copepod groups in spring was very close to that of decapods.

5 The multiple linear regression analysis performed for major copepod genera was also significant ($R^2 = 0.98$, $P < 0.001$ for $\delta^{13}$C; $R^2 = 0.85$, $P < 0.001$ for $\delta^{15}$N; see patterns in Fig. 4B). Their different $\delta^{15}$N values clearly showed that there were two to three different trophic positions in the mixed copepod assemblages. The values of *Tortanus* (13.8 ± 2.1‰) and *Labidocera* (13.4 ± 3.6‰) were the highest among taxa, followed by that of *Sinocalanus* (11.3 ± 2.0‰), and were higher than those of *Paracalanus* (10.0 ± 2.3‰), *Acartia* (8.8 ± 1.5‰), and *Corycaeus* (9.4 ± 1.4‰). With the exception of brackish water

10 calanoids, their mean $\delta^{13}$C values were higher than those of putative food resources (nanoplankton and microplankton; Fig. 4A). In contrast, brackish water calanoids had more negative $\delta^{13}$C values than those of food sources. The brackish water genus *Sinocalanus* followed the pattern of brackish water calanoids. Interestingly, *Paracalanus* of the dominant marine calanoids also had a low $\delta^{13}$C value similar to brackish water calanoids.

Considering nanoplankton as the trophic baseline (TL = 1), the TL value of microplankton was calculated to be 0.7 times

15 higher than that of nanoplankton (Fig. 5). As a whole assemblage balanced from different feeding behaviors, as indicated by the standard errors, copepods occupied a 1.2 level higher TL than that of nanoplankton, indicating herbivory (here, herbivory means a trophic level of 2) on nanoplankton with slight omnivory (TL = 2–3) on other dietary sources. The TLs of three major groups (marine calanoids, brackish water calanoids, and cyclopoids) were all similar to the bulk copepod assemblage with mean levels slightly higher than 2. The mean TL value of decapods was also close to 2. In contrast, the mean TL of

20 harpacticoids was very low, reflecting their low $\delta^{15}$N values. Among marine calanoids, the mean TLs of *Acartia* and *Paracalanus* were low (~ 2), whereas the levels of *Tortanus* and *Labidocera* were high (> 3). For two brackish water calanoids, the mean TL of *Sinocalanus* (2.9 ± 0.7) was close to that of carnivores (TL ≥ 3), while that of *Pseudodiaptomus* (1 ± 0.5) is indicative of their herbivorous and/or detritivorous characteristics.

Based on TLs, average trophic enrichments of 3.9‰ and 2.4‰ for $\delta^{15}$N values in nanoplankton and microplankton,

25 respectively, were recorded for all measured zooplankton groups (Fig. 6A, B). The mean $^{15}$N enrichments of the copepod assemblage were estimated to be 3.5‰ and 1.9‰ for nanoplankton and microplankton, respectively. The enrichment values for nanoplankton feeding on marine and brackish water calanoids and cyclopoids, as well as for the genera *Paracalanus* and *Corycaeus* were close to the average value. The trophic enrichments of *Acartia* feeding on nanoplankton (3.1 ± 1.7‰) and microplankton (1.3 ± 2.0‰) were lower than for other marine calanoids. Three higher TL genera *Tortanus*, *Labidocera*, and

30 *Sinocalanus* had high enrichments from nanoplankton and microplankton, assuming that they fed on them. In contrast, the brackish water calanoid genus *Pseudodiaptomus* had low $^{15}$N enrichment. The $^{13}$C enrichments for all groups were on average 0.6‰ and 0.3‰ when feeding on nanoplankton and microplankton, respectively (Fig. 6C, D). Decapods had relatively low enrichments of both $^{15}$N and $^{13}$C compared with copepods.





### 3.4 Contribution of size-fractionated POM to copepod diets

The Bayesian mixing model calculations showed that the contributions of different sizes of POM to copepod diets varied significantly with season (Student's $t$-test, $P < 0.001$ for all cases; Fig. 7). Size-selective feeding phenomena were particularly apparent in winter (Fig. 7A) and summer (Fig. 7C). Mean contributions of microplankton accounted for about

5 two-thirds of their assimilated diets at all stations in winter and summer, and were almost equal to that of nanoplankton in spring and autumn (Fig. 7B, D). Furthermore, the proportions of the two size fractions of POM averaged from all four seasons contributing to copepod diets at different stations were also distinctly different except for station 8 (Fig. 8). The mean contributions of microplankton to the copepod diets increased gradually from the river mouth up to a peak ($0.81 \pm 0.11$) at the middle part of the bay. Then, the proportion declined gradually to a trough ($0.31 \pm 0.18$) at the deep-bay channel. The

10 proportion then rebounded to a high level again at the bay mouth station.

Major groups of copepods and decapods (spring data available) showed contrasting size-selective feeding behaviors (Fig. 9). Marine calanoids typically preferred feeding on larger particles, with a contributing proportion of $0.63 \pm 0.03$ (range: 0.51–0.78) for microplankton (Fig. 9A). Cyclopoids also preferred microplankton but their reliance on this diet was lower than that of marine calanoids (Fig. 9C). Harpacticoids had a more apparent size-selective feeding behavior and merely fed on

nanoplankton (extremely low reliance of < 0.11; Fig. 9D). In contrast, brackish water calanoids had a broad feeding size spectrum as shown by almost overlapping dependences on nanoplankton ($0.53 \pm 0.05$) and microplankton. Decapods were identified to feed on plankton with little size selectivity, with almost equal reliance on both items.

For some dominant and frequently occurring omnivorous copepod genera (e.g., *Acartia*, *Paracalanus*, *Centropages*, *Pseudodiaptomus*, and *Corycaeus*), the Bayesian mixing model calculations showed different patterns in their dependence on

nanoplankton vs microplankton (Fig. 10). Because the statistically insignificant results from the multiple linear regressions for *Centropages* prevented our estimation of the $\delta^{15}N$ values of this genus, we only used the $\delta^{13}C$ value and the mean values of marine calanoids as enrichment factors to apply the mixing model to this genus. While *Acartia* significantly preferred large to small particles (Student's $t$-test, $P < 0.001$) with a reliance of $0.86 \pm 0.03$ on microplankton (Fig. 10A), *Paracalanus* and *Pseudodiaptomus* apparently preferred small particles (Student's $t$-test, $P < 0.001$) (Fig. 10B, C). In contrast, *Corycaeus*

and *Centropages* were typically omnivorous with a reduced size selectivity (Fig. 10D, E).

The dietary compositions of three carnivorous genera (*Tortanus*, *Labidocera*, and *Sinocalanus*) were slightly different. They all showed only negligible reliance on the two size fractions of POM. *Tortanus* frequently co-occurred with many other copepods and its diet was composed primarily of *Acartia* ($0.19 \pm 0.10$), *Paracalanus* ($0.23 \pm 0.08$), brackish water calanoids ($0.22 \pm 0.13$), cyclopoids ($0.21 \pm 0.12$), and even harpacticoids ($0.11 \pm 0.08$) (Fig. 11A). Because *Labidocera* is a surface

water species, we did not consider harpacticoids to be one of its food sources. Here, *Labidocera* and either cyclopoids or brackish water calanoids did not co-occur so that we considered skipping both groups from being feasible candidates for their food sources. The results showed that *Labidocera* exclusively preferred *Acartia*. The preferred dietary sources of the





brackish water calanoid genus *Sinocalanus* were estimated to be *Paracalanus* (0.63 ± 0.08)*, Cyclopoids* (0.18 ± 0.10), and *Acartia* (0.13 ± 0.08).

## 4 Discussion

### 4.1 Variability of $\delta^{13}$C and $\delta^{15}$N values of plankton with time and space

We found that seasonal variations and the spatial heterogeneity of copepod $\delta^{13}$C and $\delta^{15}$N values in Gwangyang Bay followed those of nanoplankton (POM < 20 μm) and microplankton (POM > 20 μm) (Figs 2 and 3). Based on the results of regression analyses, we found that the variability of copepod isotopic values was influenced by salinity (spatial variations), temperature (temporal variations), and isotopic values of food sources (both spatial and temporal variations; Fig. 3). In general, spatial variations were much more pronounced because of the effect of river input and thereby riverine carbon in

different salinity regimes. More negative values of three plankton groups (nanoplankton, microplankton, and copepods) were measured near the river mouth, and then the values increased progressively to the mouth of the estuary, indicating an apparently decreasing effect of river runoff and thus the uptake of carbon derived from river-borne terrestrial organic matter. These results are consistent with other studies in estuarine environments (Cifuentes et al., 1988; Matson and Brinson, 1990; Thornton and McManus, 1994; Deegan and Garritt, 1997; Fry, 2002). Such spatial distribution patterns have also been found

for other primary producers such as seagrasses (reviewed by Hemminga and Mateo, 1996), macroalgae (Lee, 2000), as well as benthic microalgae (Kang et al., 2003), and the pattern will further propagate to consumers such as fish (Melville and Connolly, 2003; Herzka, 2005), oysters (Fry, 2002), mollusks (Antonio et al., 2010), and other benthic macro-invertebrates (Choy et al., 2008).

Seasonal successions of $\delta^{13}$C values were also apparent, probably because of high river input in spring, elevated productivity

in summer, and species successions in autumn. When the wet season started in spring and phytoplankton started to bloom, river input lowered the $\delta^{13}$C values. The values increased again in summer because of the persistence of phytoplankton bloom (a low fractionation effect because of source limitation) and elevated productivity in summer. Although both river discharge and input of light carbon were low in autumn, the observed $\delta^{13}$C values were low. This was probably because of the lack of a heavy carbon pool from microbial respiration and species succession (Rau et al., 1990). During the post-bloom

period in autumn, phytoplankton show low productivity and Chl *a* concentrations, with low abundance of diatoms but a dominance of flagellates (Baek et al., 2015). Flagellates are known to have more negative $\delta^{13}$C values than those of diatoms arising from different fractionation effects (Gearing et al., 1984; Cifuentes et al., 1988; Rolff, 2000).

The $\delta^{15}$N variability of three major plankton groups was relatively complex spatially. Spatial trends in the nanoplankton $\delta^{15}$N values can be explained by three distinct distribution patterns. The first pattern found in the river mouth area exemplifies a

30 declining trend expected by mixing of freshwater planktonic materials, which grew up in water with high levels of dissolved inorganic nitrogen. The second pattern found in the middle part of the bay, in which nitrate inputs were much reduced while





the concentrations of ammonia increased (Kwon et al., 2004; own data not shown), characterizes an increase in $\delta^{15}N$ values in association with high Chl $a$ concentrations. Fractionations by autotrophic assimilation and bacterial utilization were the most likely source of the $^{15}N$-enriched ammonia in nutrient pools of the middle of the bay (Cifuentes et al., 1988). The elevated POM $\delta^{15}N$ values in the middle of the bay may be explained by $^{15}N$-enriched ammonia remaining after algal uptake

in the river mouth channel (Sato et al., 2006). The input of sewage-derived $^{15}N$-enriched ammonia (domestic sewage and livestock waste) cannot be ruled out as an explanation for the increasing $\delta^{15}N$ values of nanoplankton. The third distribution pattern represents declining $\delta^{15}N$ values toward the offshore bay mouth in association with a reduction in the supply of $^{15}N$-enriched nutrients from terrestrial sewage. Furthermore, the fractionation effect of phytoplankton will be reduced when phytoplankton became nitrogen-limited and take up nitrogen with little fractionation (Cifuentes et al., 1988; Fogel and

Cifuentes, 1993; Granger et al., 2004). As indicated by regression analyses between the distribution of nanoplankton $\delta^{15}N$ and environmental factors, significant increases in the nanoplankton $\delta^{15}N$ values depend on ammonia (GAM, $F = 4.2$, $P = 0.029$) and Chl $a$ (GAM, $F = 3.8$, $P = 0.044$), further supporting our explanation. In addition, temperature might be another important factor that altered the seasonal distribution of nanoplankton $\delta^{15}N$ values (GAM, $F = -5.5$, $P = 0.013$), decreasing the values in summer to autumn. High temperatures probably enhance the assimilation of phytoplankton with low

fractionation effects (Barnes et al., 2007).

The microplankton $\delta^{15}N$ values were stepwise elevated by environmental factors including temperature (GAM, $F = 5.0$, $P = 0.015$), salinity (GAM, $F = 5.0$, $P = 0.031$), ammonia (GAM, $F = 4.5$, $P = 0.031$), and nitrate (GAM, $F = 7.8$, $P = 0.010$). Similar to nanoplankton, regression analysis also showed that the microplankton $\delta^{15}N$ values increased significantly with increasing Chl $a$ concentrations (GAM, $F = 4.2$, $P = 0.043$). There are two possible mechanisms for this pattern. The first is

that higher phytoplankton abundance will result in a $^{15}N$-enriched nutrient pool because of fractionation during nutrient assimilation (Kang et al., 2009). The second is that higher phytoplankton abundance will help flourish microzooplankton such as ciliates and heterotrophic dinoflagellates, which prey on phytoplankton and thereby contain higher $\delta^{15}N$ values thanks to trophic enrichment (Sommer and Sommer, 2004). Nitrate was important for microplankton, indicative of the role of diatoms (preferring nitrate) in controlling the variation in microplankton $\delta^{15}N$ values, whereas nanoflagellates (preferring

ammonia) probably controlled the variation in nanoplankton $\delta^{15}N$ values.

As indicated by GAM analysis, the seasonality of copepod $\delta^{15}N$ values was primarily enhanced by temperature, which probably caused an elevated fractionation effect during the rapid assimilation of copepods. Although the regression relationship between larger POM and copepods was significant, the patterns were somewhat decoupled, as they were primarily observed in spring and autumn. This kind of decoupling has also been reported in the open ocean (Montoya et al.,

2002), where the transfer of nitrogen from primary producers to zooplankton is weak. A time lag in zooplankton development might cause the mismatch of zooplankton to $^{15}N$-enriched POM at the initial stage of nutrient supplies. Indeed, here we found that the high $\delta^{15}N$ values of copepods were primarily observed in summer, while the corresponding $\delta^{15}N$ values of POM started to increase from the winter, and phytoplankton blooming occurred in the spring.



## 4.2 Trophodynamics and trophic enrichments of copepods

The variability of copepod isotopic values in Gwangyang Bay suggests that the TLs of the copepod assemblage are highly dynamic. Because of different feeding behaviors and fractionation effects of copepods, the variability of trophic positions of copepod assemblage depends on the overall composition of species and is determined by dominant species. Direct

measurements of copepod isotopic values for species levels have been poorly conducted in the literature, although there are still clear patterns in existing reports. In the Southern Ocean copepods, the known carnivores *Euchaeta* and *Heterorhabdus* had high $\delta^{15}$N values, while the acknowledged omnivores *Calanoides* and *Metridia* were intermediate in position, and *Rhincalanus* had the lowest values (Schmidt et al., 2003). A mesocosm study found that the $\delta^{15}$N values were increasingly higher in the order *Temora < Pseudocalanus < Centropages*, suggesting an increase of carnivory in the same manner

(Sommer et al., 2005). The trophic positions of primary consumers (*Oithona* and *Neocalanus*) and secondary consumers (*Pleuromamma* and *Euchaeta*) in the North Pacific Subtropical Gyre are estimated to be 2.1 and 2.9, respectively (Hannides et al., 2009). Furthermore, Kürten et al. (2011) reported that the relative trophic positions of zooplankton in the North Sea were high when the assemblage was mainly composed of predators such as *Sagitta* and *Calanus*, but low when the assemblage was dominated by *Pseudocalanus* and zoea larvae.

Our study has demonstrated the trophodynamics of estuarine copepods using multiple linear mixing model analysis based on the values of bulk samples and percentages in total biomass, by which the results of estimated $\delta^{13}$C and $\delta^{15}$N values were both significant ($P < 0.01$). The estimated $\delta^{13}$C values varied greatly up to 9.4‰ among groups (from the lowest for brackish water calanoids to the highest for harpacticoids) and 15.3‰ among genera (from the lowest for *Paracalanus* to the highest for *Labidocera*), respectively (Fig. 4). The estimated $\delta^{15}$N values also varied somewhat up to 3.6‰ among groups, indicating

one TL difference among groups if we consider 3–4‰ trophic enrichment of $\delta^{15}$N between two adjacent TLs (Post, 2002; Michener and Kaufman, 2007). The trophic enrichments calculated by the differences from the basal food sources indicate that the overall enrichments of copepods were around 3.5‰ and 1.9‰ from nanoplankton and microplankton, respectively (Fig. 6). Our estimation shows that calanoids (both marine and brackish water types) and cyclopoids occupy a similar trophic niche (i.e., similar mean $\delta^{15}$N values; Figs 4A, 5, and 6A, B), but have contrasting food sources (i.e., different $\delta^{13}$C values;

Fig. 4A). Calanoids in Gwangyang Bay are more diverse than cyclopoids. The isotopic values of calanoid copepods indicate a mixture of different genera including those with high $\delta^{15}$N values (*Sinocalanus*, *Tortanus,* and *Labidocera*) and with lower $\delta^{15}$N values (*Paracalanus* and *Acartia*; Fig. 4B). Consequently, the three major groups (marine and brackish water calanoids, and cyclopoids) as well as the mixture of all copepod groups were estimated to be on average one TL higher than the nanoplankton base. However, we do not necessarily conclude that they are herbivores because the nanoplankton studied here

represent POM with a size range of 2–20 μm, which may include ciliates, heterotrophic nanoflagellates, and heterotrophic dinoflagellates. Instead, all assemblages mentioned above might be omnivorous with varying levels of relative trophic positions depending on dominant species.





Among calanoids, brackish water species had significantly lower $\delta^{13}C$ values than marine species (except *Paracalanus*), indicative of an apparent effect of riverine carbon sources on brackish species and *Paracalanus* through the food web (Fig. 4). The two brackish water species *Sinocalanus tenellus* and *Pseudodiaptomus koreanus* had contrasting TL values (Fig. 5), indicating a mixture of brackish water calanoids being close to omnivory with a broad feeding size spectrum. However, as

the two species are dominant in different seasons (Table 1), the TLs of the copepod assemblage at a specific condition will become relatively carnivorous (*Sinocalanus* dominating) or detritivorous (*Pseudodiaptomus* dominating). In contrast, among marine calanoids, *Paracalanus* (*P. parvus* and *P. aculeatus*) and *Acartia* (*A. omorii*, *A. ohtsukai*, and *A. erythraea*) were the two most common genera in Gwangyang Bay throughout the year (Table 1). *P. parvus* is an important small species (body length ≤ 1 mm) that is widely distributed in coastal and estuarine waters worldwide (Turner, 2004). Our results showed that,

similar to brackish water calanoids, this species was greatly influenced by $^{13}C$-depleted dietary sources, dominating both brackish stations in winter and saline stations in autumn (Table 1). This result indicates that *Paracalanus* was well adapted to fluctuating estuarine environments by feeding on prey originating from freshwater or prey that depends on riverine carbon sources. *Acartia* had a similar trophic niche to *Paracalanus* (Figs 4B and 5), but their dietary sources differed as estimated from their $\delta^{13}C$ values. *Acartia* is also a widely distributed genus, with a switching feeding behavior in response to the status

of food composition (Kiøboe et al., 1996; Rollwagen-Bollens and Penry, 2003; Chen et al., 2013). The isotopic values of *Acartia* were similar to those of the assemblage of marine calanoids, indicating that this genus is omnivorous, as typical of marine calanoids. Conversely, two marine calanoid genera, *Tortanus* (*T. dextrilobatus* and *T. forcipatus*) and *Labidocera* (*L. euchaeta* and *L. rotunda*), were primarily carnivorous as indicated by their $\delta^{15}N$ values (Figs 4B and 5). These estimated results are consistent with the former experimental tests and field investigations (Ambler and Frost, 1974; Landry, 1978;

Conley and Turner, 1985; Hooff and Bollens, 2004). Cyclopoids (primarily *Corycaeus affinis*) dominated copepod assemblages in spring and autumn at the middle part and deep-bay channel of the bay (Table 1). Our data reveal that *Corycaeus* is primarily omnivorous, being one TL higher than nanoplankton (Fig. 5) and prefers feeding on $^{13}C$-enriched dietary sources (Fig. 4). This result seems inconsistent with previous reports that the *Corycaeus* genus is carnivorous (Gophen and Harris, 1981; Landry et al., 1985; Turner, 1986).

Isotopic values of microplankton indicate that they are roughly a half TL value higher than nanoplankton. Considering that the sizes of most ciliates and heterotrophic dinoflagellates primarily fall within this size spectrum (20–200 μm), this result suggests an omnivorous trend among the mixed microplankton groups. Decapod larvae, measured from spring samples, had a similar but slightly lower TL compared with copepods. The ability for both herbivory and omnivory allows them to gain energy from primary producers and microorganisms, even detritus and fecal pellets (Anger, 2001; and see references therein).

We found that there were similar contributing proportions of the two size fractions of POM to the diet of decapod larvae, indicating that they fed on POM without size selectivity. Similarly, although measured only in winter samples, the benthic copepod group harpacticoids, represented by the species *Euterpina acutfrons,* also differed from calanoids and cyclopoids with low $\delta^{15}N$ values. The TL of harpacticoids estimated from this approach was somewhat misleading because of their





unexpectedly low $\delta^{15}$N values, which probably reflect feeding on detritus or dead organisms that are depleted in $^{15}$N (Sautour and Castel, 1993). This might have arisen from a bias in our assumption of equal body mass among different calanoid genera.

**4.3 Selective feeding of copepods**

Feeding preferences of different groups or genera on two size fractions of POM are of particular importance to explain seasonal and spatial patterns of community trophic niches, and in turn will predict the impacts of the grazer community on lower TLs including phytoplankton and microzooplankton. Because not all possible food sources, such as bacteria, picoplankton, fecal pellets, and dead detritus, were investigated, our Bayesian mixing model calculations might have led to some biased results. Nevertheless, the model results might provide an estimation on what size fractions of dietary sources the grazer community ingest and assimilate. In general, our results highlight that the copepod assemblages have size-selective feeding behaviors, and that these vary with season and space (Figs 7 and 8) depending on the feeding preference of dominant species (Figs 9–11).

The whole copepod assemblage assimilated two-thirds of its food requirement from microplankton in winter and summer, but they fed equally on both size fractions of POM in spring and autumn (Fig. 7). Copepods in winter and summer were primarily dominated by marine calanoids, especially by the genus *Acartia* (Table 1). Based on the model results for major copepod groups and genera, marine calanoids, especially *Acartia*, preferred feeding on large particles (Figs 9A and 10A), causing the selective feeding of the bulk copepod assemblage on such particles. Such a size-selective feeding preference has been widely reported in many field investigations (Liu and Dagg, 2005; Jang et al., 2010; Chen et al., 2017). Calanoid copepods, dominated by *Acartia*, are reported to prefer feeding on phytoplankton larger than 20 μm in the coastal water adjacent to the present study area (Jang et al., 2010). Another marine calanoid genus, *Paracalanus*, preferred feeding on nanoplankton (Fig. 9B), suggesting that the filtering efficiency, with which the grazers' feeding appendage retains particles, differs between these two genera. Because of such different feeding preferences, they dominated in different stations or seasons with differing food conditions (Table 1) and frequently co-occurred with little overlap of preferred food particle sizes.

In contrast to marine calanoids (except for *Paracalanus)*, brackish water calanoids and cyclopoids, especially the species *Corycaeus affinis*, dominated the copepod assemblage in spring and autumn in Gwangyang Bay (Table 1). The model results indicated that the size selectivity of brackish water calanoids and cyclopoids is less pronounced (Fig. 9B, C), causing the copepod assemblage to feed more on small prey, resulting in roughly equal contributions of nano- and microplankton to the diet of the copepod assemblage in spring and autumn. While the size selectivity of the brackish water genus *Pseudodiaptomus* was apparently for nanoplankton, the selectivity of *Corycaeus* was unclear (Fig. 10C, D). The omnivorous feeding behavior (discussed above) and broad feeding size spectrum of *Corycaeus* are likely to make this genus widely distributed in all the world's oceans (Turner, 2004) and frequently dominate in copepod assemblages (this study). To our knowledge, the feeding habit of *Pseudodiaptomus koreanus* is unknown in the current literature, whereas some field studies





suggest that estuarine *Pseudodiaptomus* flourishes by feeding on small phytoplankton cells (< 20 μm) (Froneman, 2004), consistent with the present results.

Although the isotopic data for harpacticoid copepods and decapod larvae were limited to only one season for each group, we still obtained a clear feeding selectivity pattern based on the Bayesian mixing model (Fig. 9D, E). Harpacticoids in winter
preferred microplankton to nanoplankton when benthic food sources were not considered. Their feeding selectivity contributed to the overall feeding preference of total copepods in this season. Decapod larvae preferred feeding on nanoplankton in spring, which was similar to the pattern of spring copepods dominated by *P. parvus*. For three carnivorous genera—*Tortanus*, *Labidocera*, and *Sinocalanus*—feeding on two size fractions of POM did not occur, based on no contribution of the two size fractions of POM to their diets (Fig. 11). The principle to select food sources to test the Bayesian
mixing model was the natural co-occurrence of predators and prey. Accordingly, while brackish water calanoids, cyclopoids, and harpacticoids were not considered as food sources for *Labidocera*, such calanoids and harpacticoids were not considered as food sources for *Sinocalanus*. The SIAR mixing model results indicated that *Tortanus* predated on diverse copepods without apparent selectivity, *Labidocera* preferred *Acartia* to *Paracalanus*, and *Sinocalanus* preferred cyclopoids to *Acartia*.

## 5 Conclusions

Here we have demonstrated the temporal and spatial variability of stable isotope ratios of copepods, which was determined by the isotopic values of two size fractions of POM, and strongly influenced by salinity (spatiality) and temperature (temporality). Such characteristics are key in understanding the biogeochemical cycles of carbon and nitrogen in Gwangyang Bay. We further used a simple linear mixing model and a Bayesian mixing model to extrapolate from the information derived from the isotopic analysis of bulk copepod samples. The model results were robust and allowed the estimation of the
relative TLs and trophic enrichment (fractionation effect) of different groups and dominant genera of copepods, as well as their diet compositions. Temporal and spatial patterns of copepod isotopic traits were further explained by size selectivity on plankton size fractions, as well as the feeding preference of dominant species. Based on such relative trophic positions and feeding preference, we can depict a simple energy flow of the Gwangyang Bay planktonic food web: from primary producers (nanoplankton) and a mixture of primary producers and herbivores (microplankton) through omnivores
(dominated by *Acartia*, *Paracalanus*, and *Corycaeus*) and detrivores (represented by *Pseudodiaptomus*, and *Euterpina*) to carnivores (dominated by *Tortanus*, *Labidocera*, and *Sinocalanus*).

**Author contribution**: M. Chen and C. K. Kang designed the experiments. D. Kim and C. K. Kang conducted field work, sample collection, and laboratory analyses. M. Chen, H. Liu and C. K. Kang developed the growth model and performed the
simulations. M. Chen, H. Liu and C. K. Kang prepared the manuscript with contributions from all co-authors.





**Acknowledgements:** This research was supported by ″Long-term change of structure and function in marine ecosystems of Korea″ funded by the Ministry of Oceans and Fisheries, Korea. M.C is also supported by China Scholarship Council (CSC: 201604180018) and National Natural Science Foundation of China (NSFC: 41306168). We kindly thank Young-Jae Lee, Hee-Yoon Kang, Changseong Kim and Jenny Seong for sampling assistance and sample analysis.

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



Table 1. Seasonal variations in basic environmental factors, including temperature (T), salinity (S), and chlorophyll $a$ ( Chl $a$), copepod abundance, and dominant species and the percentage (%) of dominant species in total copepods at the 9 stations in the Gwangyang Bay system

| Seasons | Stations | T (°C) | S (psu) | Chl $a$ ($\mu$g l$^{-1}$) | Copepod Abundance (ind. m$^{-3}$) | Dominant species | % of dominant species |
|---|---|---|---|---|---|---|---|
| Winter February 2015 | 1 | 7.4 | 1.4 | 1.2 | 65 | *Acartia hudsonica* | 45 |
| | 2 | 9.1 | 8.0 | 2.2 | 551 | *Acartia hudsonica* | 89 |
| | 3 | 9.3 | 23.7 | 2.3 | 944 | *Acartia omorii* | 33 |
| | 4 | 8.5 | 28.2 | 1.6 | 1614 | *Acartia omorii* | 46 |
| | 5 | 8.4 | 29.9 | 1.0 | 2888 | *Acartia omorii* | 49 |
| | 6 | 9.1 | 27.4 | 0.8 | 2123 | *Acartia omorii* | 32 |
| | 7 | 8.7 | 26.9 | 1.3 | 1673 | *Paraclanus parvus* | 41 |
| | 8 | 9.0 | 26.9 | 1.1 | 2159 | *Paraclanus parvus* | 30 |
| | 9 | 9.0 | 26.1 | 1.0 | 2690 | *Paraclanus parvus* | 35 |
| Spring May 2015 | 1 | 19.8 | 0.0 | 5.0 | 2265 | *Pseudodiaptomus koreanus* | 88 |
| | 2 | 19.8 | 4.7 | 2.0 | 175 | *Pseudodiaptomus koreanus*; *Tortanus dextrilobatus* | 46 |
| | 3 | 19.0 | 11.2 | 0.9 | 324 | *Acartia omorii* | 53 |
| | 4 | 17.4 | 27.2 | 6.2 | 326 | *Corycaeus affinis* | 38 |
| | 5 | 17.0 | 30.1 | 2.2 | 266 | *Corycaeus affinis* | 52 |
| | 6 | 17.0 | 32.2 | 6.8 | 358 | *Corycaeus affinis*; *Calanus sinicus* | 41 |
| | 7 | 18.0 | 32.7 | 5.3 | 148 | *Corycaeus affinis* | 73 |
| | 8 | 16.5 | 32.9 | 3.8 | 139 | *Corycaeus affinis* | 41 |
| | 9 | 16.5 | 32.8 | 2.7 | 150 | *Acartia omorii* | 81 |
| Summer August 2015 | 1 | 26.8 | 0.4 | 1.0 | 53 | *Tortanus dextrilobatus* | 79 |
| | 2 | 27.4 | 10.6 | 4.3 | 3220 | *Tortanus dextrilobatus* | 58 |
| | 3 | 27.1 | 20.5 | 4.5 | 784 | *Acartia ohtuskai* | 71 |
| | 4 | 25.8 | 28.8 | 1.6 | 1401 | *Acartia ohtuskai* | 62 |
| | 5 | 23.7 | 32.2 | 2.8 | 366 | *Acartia ohtuskai* | 37 |
| | 6 | 23.9 | 32.2 | 2.9 | 129 | *Acartia erythraea* | 67 |
| | 7 | 24.1 | 32.3 | 2.3 | 79 | *Labidocera rotunda* | 60 |
| | 8 | 24.5 | 32.4 | 1.6 | 124 | *Acartia erythraea* | 93 |
| | 9 | 24.2 | 32.5 | 2.4 | 81 | *Acartia erythraea* | 55 |
| Autumn November 2015 | 1 | 8.8 | 0.0 | 0.2 | 17 | *Sinocalanus tellenus* | 78 |
| | 2 | 9.9 | 4.7 | 0.1 | 22 | *Paracalanus parvus* | 32 |
| | 3 | 11.3 | 15.0 | 0.5 | 33 | *Paracalanus parvus* | 32 |
| | 4 | 12.1 | 20.9 | 0.4 | 32 | *Corycaeus affinis* | 65 |
| | 5 | 15.4 | 31.3 | 0.4 | 18 | *Corycaeus affinis* | 62 |
| | 6 | 14.6 | 31.3 | 0.4 | 41 | *Corycaeus affinis* | 55 |
| | 7 | 14.7 | 31.8 | 0.9 | 113 | *Corycaeus affinis* | 71 |
| | 8 | 14.8 | 32.3 | 1.1 | 118 | *Corycaeus affinis* | 30 |
| | 9 | 14.2 | 32.1 | 0.4 | 23 | *Corycaeus affinis* | 56 |





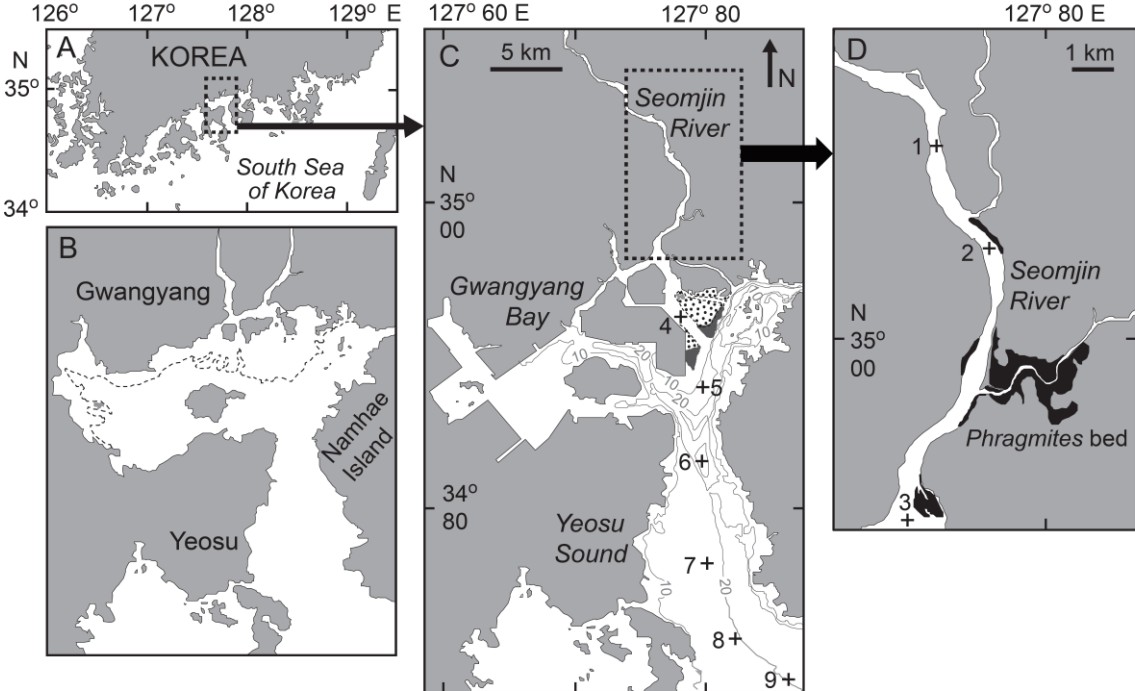

**Figure 1: Map showing the location of Gwangyang Bay (A), the appearance of the bay before the reclamation of tidal flats in 1982 (B), the sampling stations in the bay (C), and in the estuarine channel (D). The broken line represents the lowest water line (B); the dotted areas show intertidal beds, the dark gray areas *Zostera* beds (C); and the darker areas *Phragmites* beds (D).**





**Figure 2: Temporal and spatial variations in plankton $\delta^{13}C$ (‰) and $\delta^{15}N$ (‰).**



**Figure 3: Partial effects of important environmental factors on the variabilities in stable isotopes.**



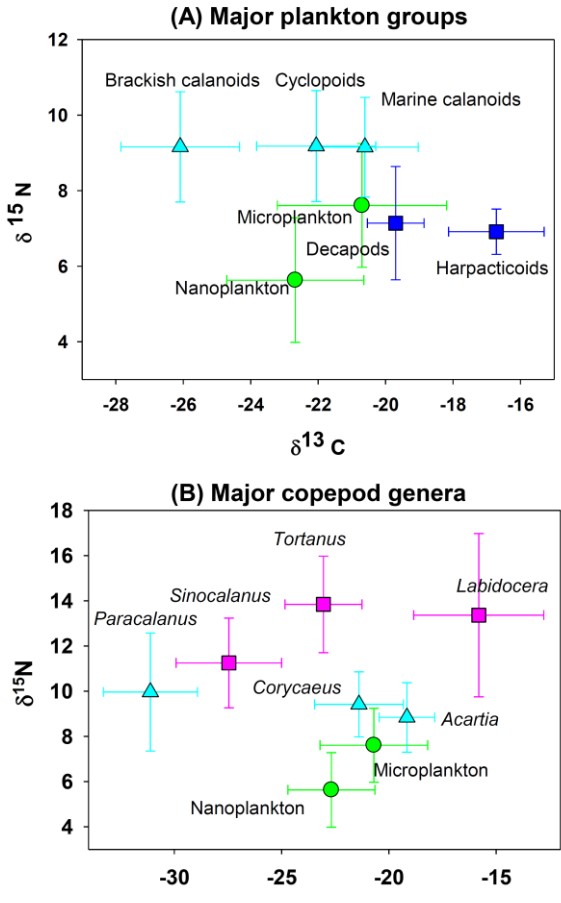

**Figure 4: Bi-plots of major plankton group and genus isotopes in Gwangyang Bay. The green circles indicate the major plankton food sources; cyan triangles indicate primary consumers; blue squares indicate omnivores or detrivores; and pink squares indicate relative carnivores.**





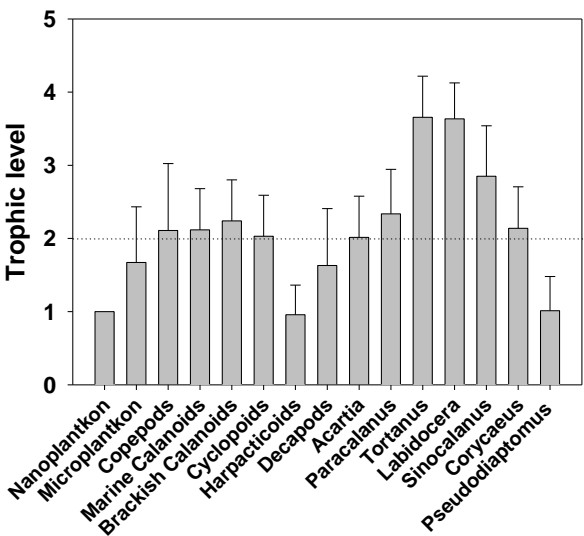

**Figure 5: Trophic levels (TLs) of different groups. Nanoplankton were set as TL = 1, while consumers' trophic levels were calculated as:** $TL = 1 + (\delta^{15}N_{consumer} - \delta^{15}N_{Nanoplakton})/3$. **The reference line indicates the herbivores relative to nanoplankton. However, nanoplankton here might not be truly primary producers as the bulk samples might include heterotrophic flagellates, dinoflagellates, and ciliates, which we could not separate from the collected samples.**



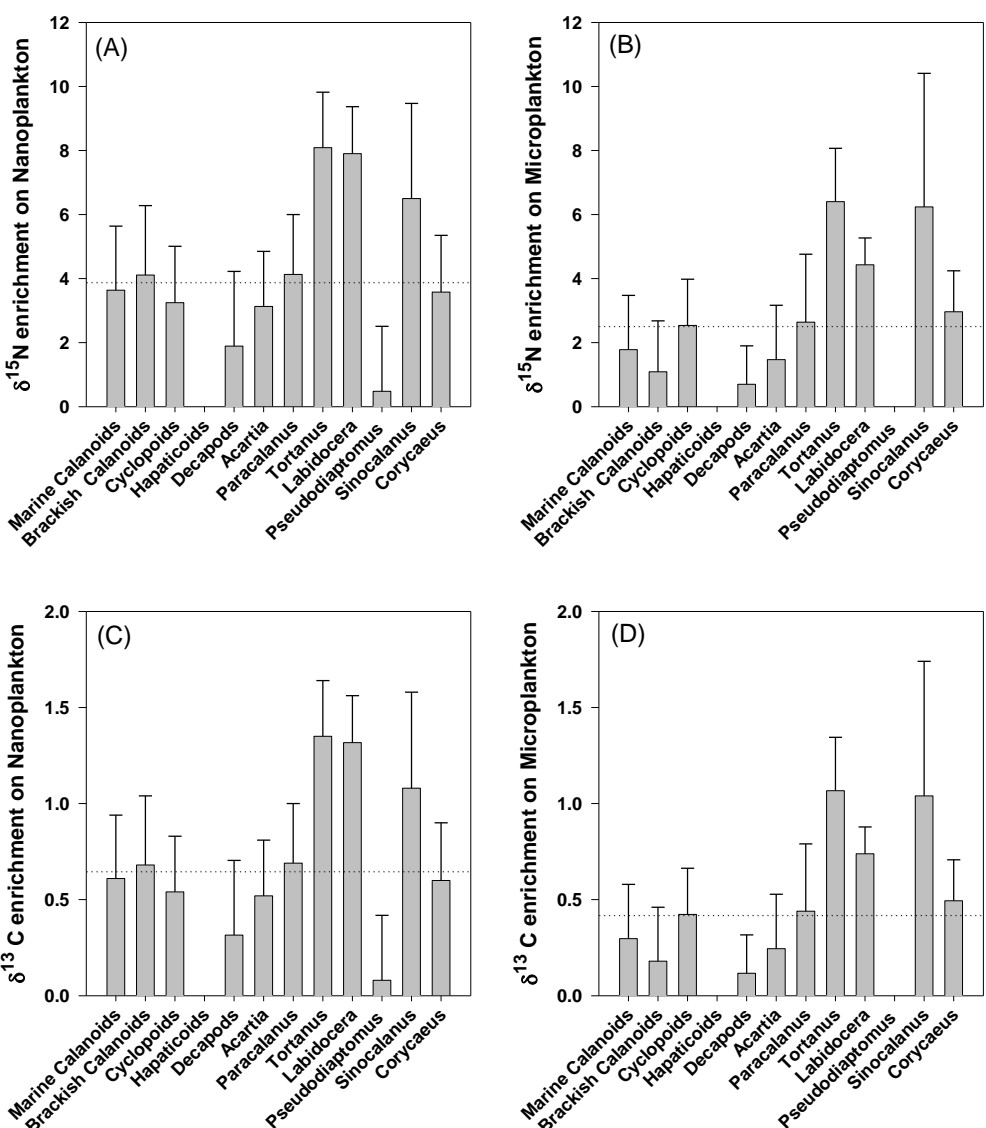

**Figure 6: Trophic enrichment (or fractionation factor) from basal food items (nanoplankton: A and C; microplankton: B and D), based on the difference of each sample's δ$^{15}$N between higher trophic level to lower trophic level; a 0.5‰ per one trophic level was used to calculate the δ$^{13}$C enrichment for each group. Reference lines indicate mean values from all groups.**



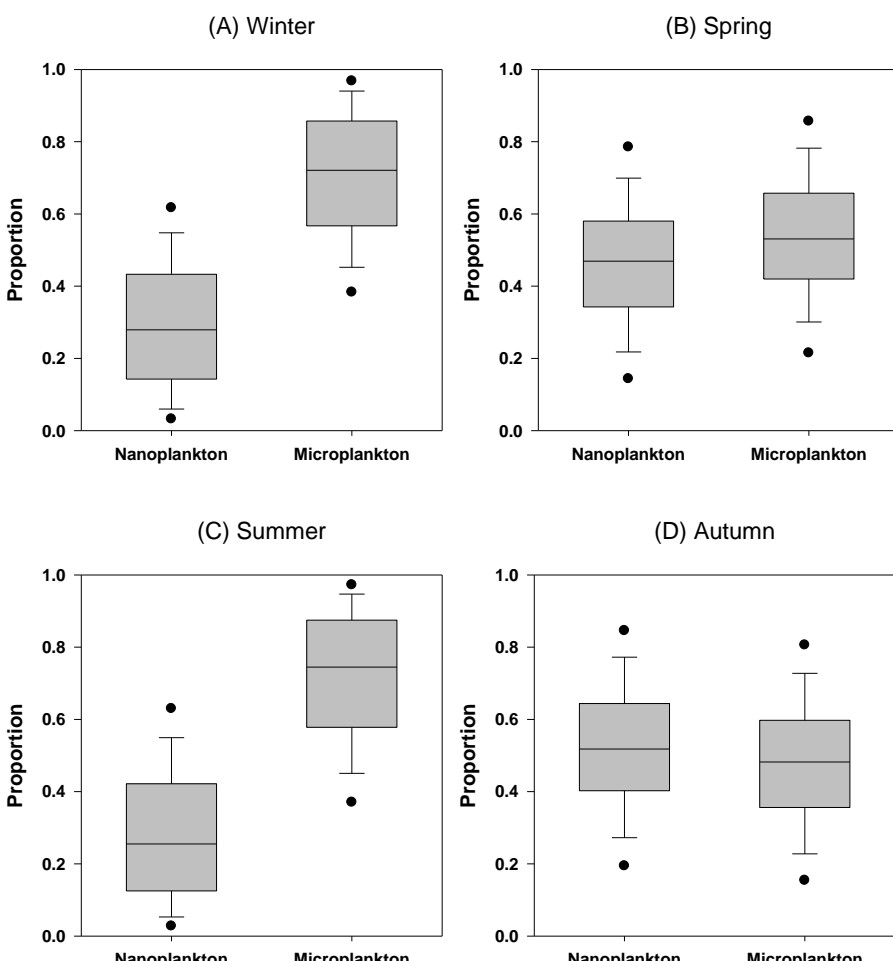

**Figure 7: Temporal variations in the contributions of size-fractionated plankton in copepod diets estimated by a Bayesian mixing model using the SIAR package in the R statistical program. Credibility intervals of 95% (dots), 75% (whiskers), and 25% (boxes) and mean values (lines in the boxes) are shown in boxplots for each season.**



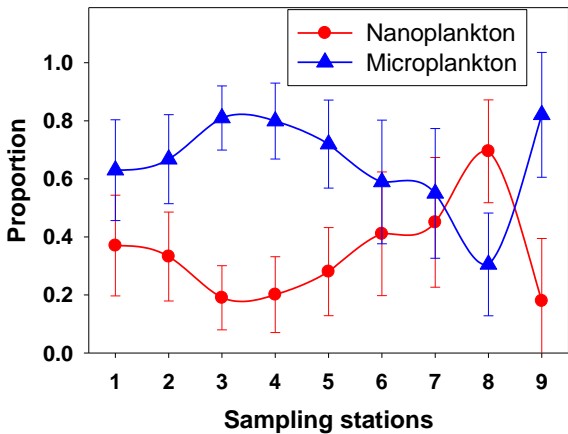

**Figure 8: Spatial variations in the contributions of size-fractionated plankton in copepod diets estimated by Bayesian mixing model using the SIAR package. Mean values ± standard deviations from all seasons are shown for each station.**





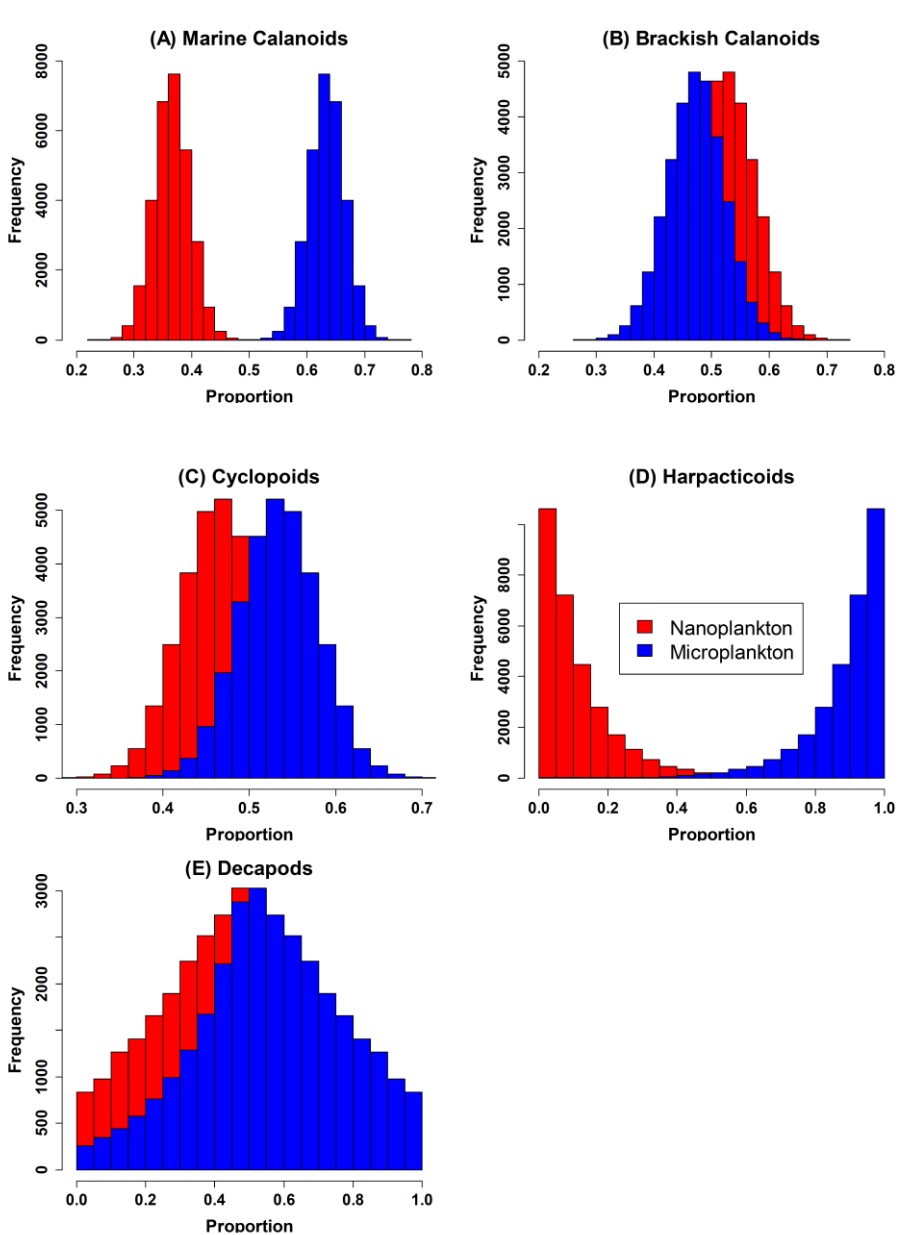

Figure 9: Comparison of the dietary compositions of major copepod groups and decapods.



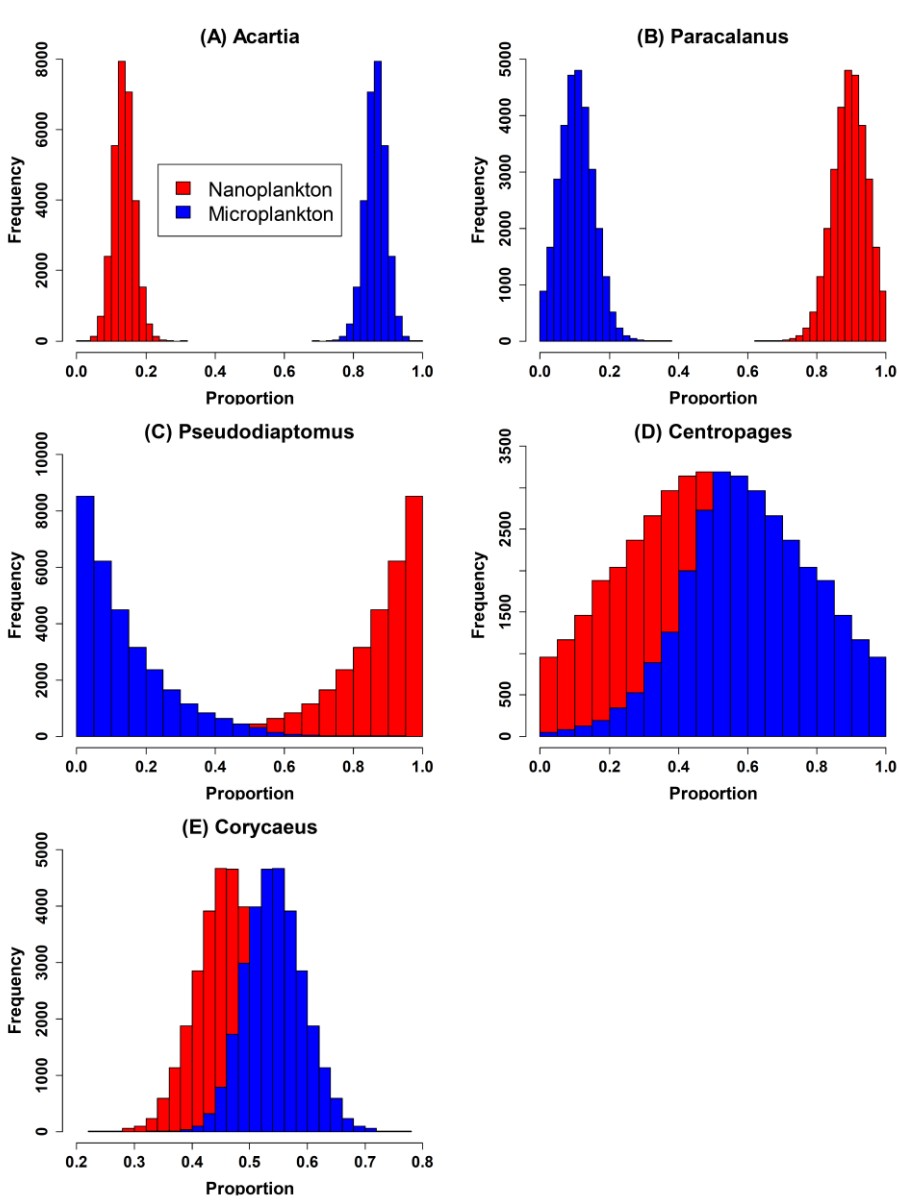

**Figure 10: Comparison of the dietary compositions of major omnivorous copepod genera.**





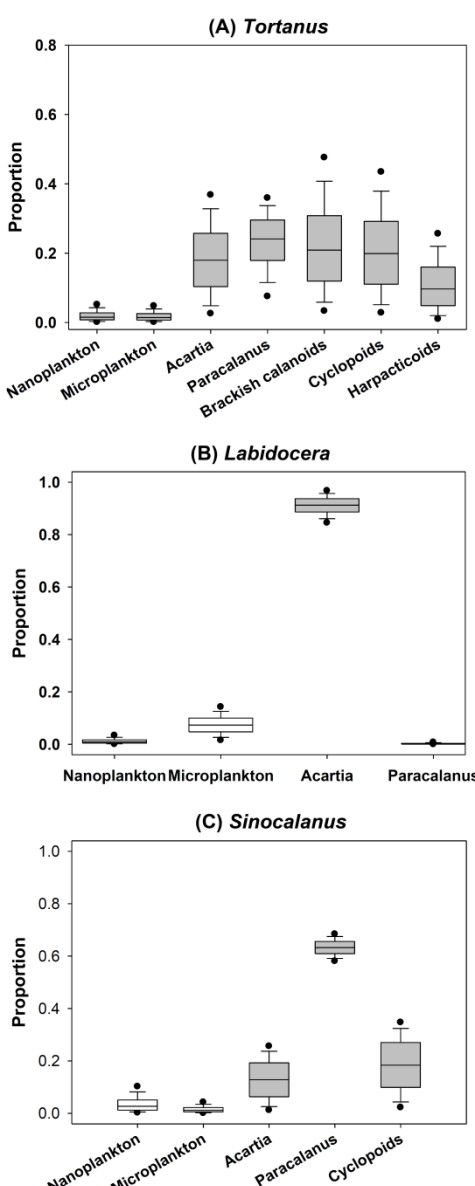

**Figure 11: Comparison of the dietary composition of major carnivorous genera. Credibility intervals of 95% (dots), 75% (whiskers), and 25% (boxes) and mean values (lines in the boxes) are shown in boxplots for each source.**

