# Peer review of "Variability in copepod trophic levels and feeding selectivity based on stable isotope analysis in Gwangyang Bay off the southern coast of Korea"

_Biogeosciences, 2017_

## Referee Comment (RC1) · Anonymous Referee #1 · 19 Oct 2017

General Comments:

The authors used stable isotope analysis to solve copepod trophism (i.e. food resources and trophic level), which is important to understand the biogeochemistry in estuarine system. The findings on copepod trophism in the manuscript (MS) will contribute for understanding pelagic food webs in the system. These are valuable and positive points in the MS.

Nevertheless, I found many doubtful points throughout the MS. The authors simplified the dynamics of copepod community by considering the most dominant copepod species only, and then applied this simplified copepod assemblage to the stable iso-

tope analysis. As a result, trophism of some copepod group, especially carnivorous copepods are still questionable.

The current method (e.g. Boyesian mixing model) and the assumption (e.g. the body mass of different genera among calanoids are the same) applied to stable isotope analysis may have some limitation to evaluate the real trophism of the copepods in the field, even though most results of copepod trophism in the MS were similar to previous reports.

Therefore, I would like to recommend the authors to include an additional explanation on a potential limitation which may occur when you apply the current method and assumption to copepod community, in the revised MS.

Specific Comments:

P6, line 20-22: The author's assumption is questionable. Calanoids consist of many genera or species with various sizes. Even though some large calanoids are not dominant in the sample in terms of abundance, some large calanoids (e.g. Calanus) can have important role in terms of biomass or volume. So, the author's assumption may not apply to a mixed copepod community with existence of both small and large copepods.

In relation to this issue, how did the authors treat copepodite stages of the copepods occurred in this study to calculate their abundance or body mass? There is no explanation in the materials and methods.

P8, line 13-19 and Table 1: There is no criterion for dominant species in Table 1. The authors listed only the most abundant copepod species by station and season. I think more than one copepod species would have contributed to copepod community in the field. Please specify a criterion and also show other copepod species if possible (in fact, the information on copepod species composition is poor and not informative in this study).

P10 line 31-33: More detailed explanation may be needed, like in the case of delta 15N in Fig. 6A and 6B.

P10 line18: There are no results for Centropages in Fig. 4, Fig. 5 and Table 1. However, the authors showed the dietary compositions of Centropages as a major omnivorous copepod genera in Fig. 10D. Why?

P11 line 29-32: The authors did not consider either cyclopoids or brackish water calanoids because those are not co-occurred with Labidocera, a surface water species. However, I believe that Labidocera has a chance to contact other preys beside Acartia and Paracalanus, such as cyclopoids and brackish water calanoids. If the authors check the copepod community in summer, not only dominant species but other subdominant species (not shown in Table 1), there are many adult and copepodite copepod species that can be a potential prey for Labidocera. So, please add potential prey in Fig. 11.

P11 line 32- P12 line 1-2: For Sinicalanus, potential prey including brackish water calanoid such as Pseudodiaptomus should be tested in Fig. 11. Also, I failed to understand that why Acartia was considered as prey for Sinocalanus in Fig. 11, considering Acartia was not dominant species in autumn in Table 1.

P14 Line23-25: I understand that calanoids (both marine and brackish water types) and cyclopoids had different delta 15N values according to Fig. 6B. However, the authors mentioned the mean value of the group was the same. Please check again.

P15 line 3: There is no result of the brackish water species, Pseudodiaptomus in Fig. 4, but in Fig. 5. Why?

P15 line 20-24: Corycaeus affinis was evaluated as omnivorous in this study, but as carnivorous in previous reports. What is a possible explanation for this difference?

P15 line 27-31: I believe decapod issue is not necessary for this study. Why did the authors include decapod results?

P15 line 31-33: There is no result of Euterpina as a genus of benthic harpacticoids in the results section, but only as harpacticoids. However, the authors mentioned Euterpina was detrivores in discussion and conclusion. In case of cyclopoids, the trophic level of cyclopoids and Corycaeus was presented separately in Fig. 4. Why?

P16 line 13-16: Even though Acartia dominated the marine calanoids in winter and summer, it is questionable to say that the bulk copepod assemblage with various species prefers large particles (microplankton; Fig. 7A and 7B). Likewise, Paracalanus also dominated the marine calanoid community in the more saline region in winter (Table 1), and Paracalanus prefers small particles (nanoplankton; Fig. 10). Paracalanus and other marine calanoids other than Acartia also may have contributed to the feeding selectivity of the bulk copepod assemblage differently.

P16 line 31: Corycaeus affinis dominated copepod community in spring and autumn, except for the river mouth. This result is inconsistent with previous reports in the same region; Corycaeus affinis was not a dominant species in spring and autumn (Kwon et al. 2001, Jang et al. 2004). I am very curious about the difference. My speculation is that horizontal net towing (0.5-1m depth) in the deeper region in this study may be responsible for potential bias of copepod composition.

(Kwon KY, Lee PG, Park C, Moon CH, Park MO. 2001. Biomass and species composition of phytoplankton and zooplankton along the salinity gradients in the Seomjin River estuary. The Sea, J Korean Soc Oceanogr, 6: 9-102 Jang MC, Jang PG, Shin K, Park DW, Chang M. 2004. Seasonal variation of zooplankton community in Gwangyang Bay. Korean J Environ Biol, 22: 11-29)

P16 line 32: The authors concluded that Pseudodiaptomus was a detrivore, feeding on small phytoplankton cells. However, recent paper (Kayfetz and Kimmerer 2017) showed that P. forbesi in San Francisco Bay is rather omnivores feeding on various kinds of preys including centric diatom, pennate diatom, diatom (7-15$\mu$m), flagellates, flagellate (7-15$\mu$m), dinoflagellate and ciliate in the laboratory.

(Kayfetz K, Kimmerer W. 2017. Abiotic and biotic controls on the copepod Pseudodiaptomus forbesi in the upper San Francisco Estuary. Mar Ecol Prog Ser, 581: 85-101)

P17 line 5-6: The authors mentioned that harpacticoids contributed to total copepod diet, preferring microplankton in winter (Fig. 7A), because harpacticoid preferred microplankton (Fig. 9D). However, harpacticoids are not a dominant group in winter (see Table 1).

P17 line 9-13: The authors used the Bayesian mixing model to estimate the relative contribution of copepods to the carnivore diets, and the prey copepods which were not occurred with predatory copepods according to Table 1 were not considered in the model processing. However, this assumption or process may brings bias when evaluate the prey copepod contribution to predators in reality. The authors did not consider some copepod prey for Labidocera and Sinocalanus, but not Tortanus in Fig. 11. I guess that Labidocera who living on surface also may contact copepods other than Acartia and Paracalanus (for example, according to Table 1, in summer Labidocera rotunda co-occurred with Tortanus as well as Acartia spp.). Therefore, the brackish calanoids and cyclopoid also need to be included in potential prey for Labiocera. The same logic can be applied to Sinocalanus. Although Sinocalanus tellenus dominated in autumn with Paracalanus and Corycaeus, only Acartia was considered as prey for Sinocalanus, but not brackish water calanoid such as Pseudodiaptomus. Please consider all potential prey for Labidocera and Sinocalanus like in the case of Tortanus in Fig. 11A.

Also, it is not clear whether the dietary composition of the carnivorous genera in Fig. 10 was for a season or for the four seasons. Please specify appropriate season for each carnivorous copepods (e.g. all season or particular season) so that we can guess the potential prey for the carnivorous copepods.

P28 Fig.4: Please indicate which genera are the brackish calanoids or marine calanoids in Fig. 5(B) and/or Fig. 5. Also, please specify whether the result of decapods or harpacticoids is for spring and/or winter samples.

P33 Fig.9: Please indicate appropriate season for each copepod group and decapods.

Technical Corrections:

P15 line 11: 'brackish stations in autumn and saline stations in winter' instead of 'brackish stations in winter and saline stations in autumn'

P 15 line 24: 'Turner, 1984' instead of 'Turner, 1986'

P16 line 20: 'Fig. 10B' instead of 'Fig. 9B' for Paracalanus

P17 line13: 'Sinocalanus preferred Paracalanus to Acartia and/or cyclopoids.' instead of 'Sinocalanus preferred cyclopoids to Acartia.'

---

## Referee Comment (RC2) · Anonymous Referee #2 · 20 Oct 2017

Biogeosciences Discuss., https://doi.org/10.5194/bg-2017-364 Chen et al., Variability in copepod trophic levels and in feeding selectivity based on stable isotope analysis in Gwangyang Bay off the southern coast of Korea

General Comments: Chen et al report seasonal and spatial variations of copepods on 13C and 15N values in a temperate estuarine system. They present a nice description of these data and use a lot of mathematical analysis models (linear mixing models, Bayesian isotopic mixing models and generalized additive models) to deeply analyze the trophic structure of plankton. I am in favor of some salient results on averaged trophic position of different copepods and contribution of two size fractions of diets.

These kinds of results are hard to be obtained by direct measure as copepod community is highly complex so that the individual samples are difficult to separate, which also claimed by the authors. Although the size-selective feeding behaviors of copepods are not new in literature, the patterns shown in this manuscript are reasonable. More important, it still provides a powerful technique to treat such investigation data that can be followed by readers and provide insight biogeochemical information about the trophic interaction between copepods and primary producers. Therefore, this is potentially a very useful paper providing important information and methods for the biogeochemical study (i.e., food resources and trophic levels) in the complex coastal ecosystem, as well as the influence of the freshwater input in an estuary.

The main shortfall of this manuscript is that it can only provide the trophic information of several major genera of copepods. Genera with low biomass or appearance frequency like Euchaeta, Calanus and Oithona, which are also popular in the world ocean cannot be treated by the same way. In addition, I would like to suggest more discussion about the uncertainty or disadvantages of these analysis models. And, reasons for some results on feeding pattern of some species are not discussed enough. For example, what are the mechanism of the feeding selectivity of the three carnivorous genera— Tortanus, Labidocera, and Sinocalanus? Finally, it will be more visual if the authors can provide a conceptual map about the planktonic food web from their conclusion, showing the relationship and the seasonal differences of the energy flow on this map as well? Overall, I recommend this manuscript for publication in Biogeosciences with minor revision. Some specific comments are indicated below.

Specific comments: 1. A reason or a reference to calculate trophic enrichment is needed. 2. L21 $\mu$m 3. How about the errors or residual (eq. 4) for Linear regression models? 4. Y-axes in Figure 5 to 11 need plural number. 5. Increase the resolution of Figure 3.

---

## Referee Comment (RC3) · Anonymous Referee #3 · 12 Nov 2017

Review of "Variability in copepod trophic levels and in feeding selectivity based on stable isotope analysis in Gwangyang Bay off the southern coast of Korea". Authors: Mianrun Chen, Dongyoung Kim, Hongbin Liu, and Chang-Keun Kang. Submitted for review for journal Biogeosciences.

The authors use an approach that combines Generalized additive models (GAMs) and multiple regressions using bulk carbon and nitrogen isotopes to address trophic relationships among zooplankton taxa and POM, along a salinity gradient in Gwangyang Bay, off the southern coast of Korea. They find significant spatial variability, somewhat coherent patterns among copepods and microplankton, and seasonal variability in both

d13C and d15N. Using an abundance-weighed regression approach, the authors estimate bulk values for each copepod genera or taxonomic grouping, and from these infer particle selectivity, diet and trophic level (TL), for each genera with significant regressions.

General comments:

I find a problem in the way the authors estimated the weight differences between cyclopoids and calanoids randomly, as well as assuming that the weight of all calanoid genera was the same. In particular, because the authors have the taxonomic information already, I suggest they do a literature review and obtain the average weight values for each of the copepod genera/species used in the study, and apply these to the bulk regressions. I believe this is especially important as the authors are trying to extrapolate significantly more results than what they measured (i.e. genera-specific isotope values from a mixed community), that the approach be as precise as possible.

In general I appreciate the effort to expand upon simple d13C and d15N bulk measurements for more detailed information on a community. However, in the case of copepods, if the authors do/did intend to investigate these relationships in detail, why not simply measure the values of individual genera? They state that too much material is required, but methodological advances these days mean that an individual Calanus female can indeed be analyzed ($\sim$60ugC, 10ugN), as 5 ugN is typically the lower limit of standard bulk analyses (and low-N methods methods have been developed to go down to $\sim$1 ugN). Cyclopoids would require greater number, but following the authors assumptions of $0.1 < x < 1$, that would be about 10 individuals. When certain problems arise, such as Paracalanus and Sinocalanus having lower d13C values than any measured prey, it would seem the authors acknowledge them, but then continue their analyses, e.g. calculate a TL (presumably based on prey that has been shown to not be consistent with their isotope values) in the same was as for the other genera. One gets the sense that by plugging it into GAMs and regression models, the error sources and magnitudes are lost. I would like to see a quantitative test of the biases inherent in this

[Figure]

Bayesian model, and how confident the authors can be that this approach is recovering the actual copepod diets. Given this approach and the number of assumptions that lie within, uncertainty relating to the model (as well as replication, independently) should be presented, discussed, and assessed explicitly with the other sources of uncertainty. This should be done with both the particle feeders and the carnivorous species, and the effects of including or excluding different species types should also be assessed.

Finally, consistent with the point I discuss above, the authors mention a 'simple energy flow' in the abstract and discussion. But I wonder if this methodological approach allows for more complex flows. The actual isotopic values were not measured, but inferred from mass balance of dominant genera, and Bayesian approaches, and the violation of the underlying assumptions was not determined. How would a more complex picture emerge? In fact, the problem of Paracalanus and Sinocalanus having lower d13C values could hint at more complexity, yet it is assumed perhaps that this is due to unmeasured food sources and then ignored.

I think if the authors address the issues posed above (and specifics below), the MS is suitable for publication.

Specific comments:

Abstract. P1 – 10. The word 'trophism' is introduced yet does not technically mean what the authors define it as (food resources and trophic levels), and is not used within the field's jargon as such either. I would prefer 'trophic structure' or 'trophic interactions', or 'trophic preferences'.

'Temperature-related' seasonal variations – The effect of temperature from season was not separated in this study, it should be simply 'seasonal variations'.

Introduction

P2-0. "With broad feeding spectra and flexible feeding strategies, the bulk copepod assemblage is omnivorous depending on dominant species or group". Omnivorous or what? Consider changing to something like 'displays varying degrees of herbivory/omnivory/carnivory, depending on dominant. . .'

P2-5. "In turn, TLs of a diverse. . .". I assume the authors here refer to the average trophic position of the assemblage, and thus should be 'TL' (singular). "Because copepods play a fundamental role in feeding on phytoplankton as primary consumers". Consider re-phrasing as 'Because copepods rely significantly on phytoplankton as prey', otherwise the expectation of this phrase is that the second half will refer to the top-down effect of copepods on phytoplankton, and not the bottom-up effect of phytoplankton on copepods. 'feeding on phytoplankton as primary consumers, so the seasonal and spatial'. Delete 'so'.

P2-15. "Therefore, the assessment of the trophic position (. . .) of copepods within a complex planktonic food web is critical in predicting the ecological relationships between predator and prey". This phrase seems redundant, isn't the study about assessing these ecological relationships? I don't understand the prediction part.

P3. 0. "In contrast, the d15N values of primary producers increase from being nutrient-sufficient (high fractionation) to nutrient-limiting (low fractionation) and are especially high in anthropogenic wastewater nitrogen inputs". Would the later simple swamp the fractionation effect? The literature on ïĄď'15N of different nutrients in the ocean (nitrate, ammonia, urea) shows ranges that are much larger than fractionation factors, e.g. these vary by about 20‰ compared to 3.4‰ of fractionation. Can you comment on how much you expect the source to vary along the river gradient?

Materials and methods

P4-25. Could you mention the average volume filtered per tow, as the net was equipped with a flow meter?

P5-5. "water samples were transported to the laboratory as soon as possible". Please give a time estimate.

P6-5. The analytical precision of 0.2‰ and 0.3‰ for d13C and d15N, respectively, seems a bit high. Could you estimate what is the lowest change in TL that you can estimate based on this instrument error?

P6.15. The weight difference between cyclopoids and calanoids was generated randomly. I don't understand why the information from the species identification was not used for this purpose. What is the error associated with this type of computation? I would really suggest the authors do a literature search of the mean weights of the difference species and genera enumerated in their samples, and use this information to estimate both cyclopoid/calanoid weights, and the weights of the different calanoid genera. If the composition has already been estimated, it makes no sense to make these assumptions that only introduce greater error into an already indirect way of estimating species stable isotope composition.

P7-20. 'fractionation factors used in the model estimation were calculated from TLs'. I don't understand this statement, it sounds like 3‰ and 0.5 ‰ were assumed (logically) and not calculated. Please clarify.

Results

The authors discuss their seasonal results in the context of 'temperature'. I would prefer to see this discussed as 'seasonal', since temperature variability within a season was not tested and hence the driver of the observed effects cannot be unequivocally stated to be temperature. Rather, they are probably a combined effect of the changes that co-occur with each season and should be stated as such.

P8-10. 'Despite insignificant spatial variability, higher Chl a concentrations generally occurred in the middle of the bay'. This is not obvious from the values in the table. Please explain in more detail or remove.

P8-25. Please give a mean value for copepod d13C as done for the groups above (nanoplankton and microplankton)

P9-0. "Overall, seasonal succession of winter-spring, spring-summer, and summer-autumn were apparent for all plankton groups". Not clear what this means. There appears to be significant overlap in values for the nanoplankton, and no clear increasing progression from winter to autumn, as increases/decreases seem to interchange.

P9-5. It isn't clear to me how the coefficients of variation are calculated. The range of d15N values encountered is less than that for d13C, although the spatial progressions are less monotonic. Please clarify in the methods how this is calculated. P9-10. The result for the microplankton is inconsistent with the figure. In the figure, the highest value for d15N is 10‰ at the bay in spring. There is no 16.2 value.

P9-15. "Copepod d15N . . . being much more consistent with the pattern of microplankton than that of nanoplankton". This seems true for the summer ïА̧d'15N values, and quite the opposite for the winter values. Regardless, there is such high variability that it is hard to tease out any clear pattern of spatial/seasonal co-variability.

P9-20. The GAM result is very interesting. Perhaps it reflects the food-web processes that affect d15N disproportionally and were not included in the GAM?

P10-20. It is not clear to me how the trophic levels of brackish copepods can be calculated, when their 13C values do not support the sampled nanoplankton and/or microplankton as their food source. I also don't understand how later in figure 6 they show up enriched, but in figure 4 they are depleted with respect to this food source. The differences between these two figures should be stated clearly as they show different results.

P10-25. "The enrichment values for nanoplankton feeding on marine and brackish water calanoids. . .". This phrase says that nanoplankton are feeding on copepods. That's not right, it should say something like 'enrichment values for marine and brackish water. . . feeding on nanoplankton'.

P11-5. I disagree with the statement (based on the figure) that "the proportions of the

two size fractions of POM averaged from all four seasons contributing to copepod diets at different stations were also distinctly different except for station 8 (Fig. 8)". It seems that the error bars overlap at station 1 (hence not different), and stations 6 and 7. I might be missing something but then it should be clarified. P11-10. Does 'spring data available' mean 'only spring shown'?

The authors discuss size-selective feeding of calanoids in the context of 'filtering efficiency', yet they are not true filter feeders, they are suspension feeders that trap and handle particles (Paffenhofer et al, 1982, Mar Bio 67:2), which has different implications for particle handling. This is an important distinction that should be observed throughout the MS.

Discussion P13-0. It seems to me that the sewage explanation deserves a bit more attention. If the authors can't rule it out it means that this could contribute substantially and swamp the other subtle processes discussed in the 15N-enriched ammonia section.

P13-5. "Furthermore, the fractionation effect of phytoplankton will be reduced when phytoplankton became nitrogen-limited and take up nitrogen with little fractionation". I am unsure that this effect could be significant in a coastal areas such as this one. Moreover, if phytoplankton reduce their fractionation, it would mean that their 15N will tend to be higher (as they choose the lighter 14N), and thus doesn't explain this decreasing trend.

P13-10. I would like to see table with the GAM results. It would be nice to have these presented first in the results, and later discussed. It would also be interesting to see the different variables tested and the ones found to be significant within this table.

P13-20. But see Gutierrez-Rodriguez et al (2014, L&O, vol:59, i5) on negligible trophic enrichment of heterotrophic protists.

P14-0. "Because of different feeding behaviors and fractionation effects of copepods,

the variability of trophic positions of copepod assemblage depends on the overall composition of species and is determined by dominant species." Change to "...the variability of the average community trophic position depends on the overall composition of species and is determined by the dominant species."

I am somewhat confused about the discussion of trophic levels of the copepods Paracalanus and Sinocalanus. The authors state that their ïАď13C values are lower than all measured food sources, which would imply that their food source has not been adequately measured. How then are these organisms included in the trophic level (TL) component of the paper? A bit of clarification on this topic would really help the reader.

P17-10. This paragraph explaining the Bayesian mixing model methods/results should be moved to the results section.

---

## Referee Comment (RC4) · Anonymous Referee #4 · 14 Nov 2017

Review of the manuscript "Variability in copepod trophic levels and in feeding selectivity based on stable isotope analysis in Gwangyang Bay off the southern coast of Korea" by Mianrun Chen, Dongyoung Kim, Hong bin Liu, Chang-Keun Kang, Biogeosciences Discuss., bg-2017-364

This manuscript provides results from seasonal and spatial variation in the stable isotopes 13C and 15N of POM and copepods along a salinity gradient in Gwangyang Bay, off the southern coast of Korea. The authors combined this information with linear mixing models, Bayesian isotopic mixing models and generalized additive models to derive a statement on food selectivity and trophic level of copepods. In general, this

manuscript is very well structured and provides valuable information on the flow of matter through the food web. Still, some concerns have to be clarified before publication.

Below are some specific comments.

Introduction 1. Page 3, line 7: Please give more information here on the usage of different N sources and enrichment factors.

2. Page 3, line 19: "highly mixed species"- Please clarify, mixed with what? 3. Page 3, line 21: Instrument sensitivity has increased and compound specific analysis (CSI) of stable isotopes in amino acids make it possible to track diets of mesozooplankton and determine their trophic position. 4. Page 3, line 21: Please give some reason why this site was chosen.

Material and Methods 5. General: why did the authors not use literature data on average weight values for each of the species investigated instead of assigning the weight to each group? 6. General: How where copepodite stages treated regarding abundance and body mass? 7. Page 4, line 15: Change to "increasing". 8. Page 4, line 16: Specify "in the middle of Gwangyang Bay. 9. Page 4: Please add information on when sampling took place- day or night? 10. Page 5, line 11: "pico- and nano- sized phytoplankton". Doesn't sampling with a mesh also include nanozooplankton like heterotorphic and mixotrophic flagellates- so it does not only comprise phytoplankton?! 11. Page 6, line 29: something is missing at the end of the sentence- "illustrated in figures?".

Results and Discussion

12. There are too many figures. Some might be moved to the supplemental section, e.g. Fig. 3, 7,8,11 13. Page 12, line 16: What is a "heavy carbon pool", give an example? 14. Page 12, line 31: Wording! Please revise "much reduced". 15. Page 13, line 15: "with low fractionation effects"- give example.

Conclusion

16. Please provide a simplified figure of the energy flow for the different seasons.

---

## Author Comment (AC1) · 13 Dec 2017

General Comments: The authors used stable isotope analysis to solve copepod trophism (i.e. food resources and trophic level), which is important to understand the biogeochemistry in estuarine system. The findings on copepod trophism in the manuscript (MS) will contribute for understanding pelagic food webs in the system. These are valuable and positive points in the MS. Nevertheless, I found many doubtful points throughout the MS. The authors simplified the dynamics of copepod community by considering the most dominant copepod species only, and then applied this simplified copepod assemblage to the stable isotope analysis. As a result, trophism of some

copepod group, especially carnivorous copepods are still questionable. The current method (e.g. Boyesian mixing model) and the assumption (e.g. the body mass of different genera among calanoids are the same) applied to stable isotope analysis may have some limitation to evaluate the real trophism of the copepods in the field, even though most results of copepod trophism in the MS were similar to previous reports. Therefore, I would like to recommend the authors to include an additional explanation on a potential limitation which may occur when you apply the current method and assumption to copepod community, in the revised MS. - Response: We thank the reviewer for his/her careful checking of the manuscript and we agree his/her general assessment of our work and are happy that he found our valuable and positive points in the MS. We agree that the current method has some limitations to study the entire complex planktonic structure. However, the reviewer may misunderstand our data set as we were not just considering the most dominant species only, whereas we considered all appearing copepod genera in statistics when we interpreted the dynamics of the entire copepod community or specific taxonomic groups like calanoids and cyclopoids. We believed that copepod genus can be grouped based on similar feeding behaviors and thus the food web structure can be simplified.

Specific Comments: P6, line 20-22: The author's assumption is questionable. Calanoids consist of many genera or species with various sizes. Even though some large calanoids are not dominant in the sample in terms of abundance, some large calanoids (e.g. Calanus) can have important role in terms of biomass or volume. So, the author's assumption may not apply to a mixed copepod community with existence of both small and large copepods. - Response: We admit that this assumption was a little bit inaccurate. Unfortunately, we didn't record the body length of our taxonomic data. As suggested by another reviewer, we will re-calculate their biomasses based on the empirical formula of biomass and body length of different copepod genus and use the ratio of body length among different genera.

In relation to this issue, how did the authors treat copepodite stages of the copepods

occurred in this study to calculate their abundance or body mass? There is no explanation in the materials and methods. - Response: Copepodites were counted but grouped together with adults.

P8, line 13-19 and Table 1: There is no criterion for dominant species in Table 1. The authors listed only the most abundant copepod species by station and season. I think more than one copepod species would have contributed to copepod community in the field. Please specify a criterion and also show other copepod species if possible (in fact, the information on copepod species composition is poor and not informative in this study). - Response: Species with abundance consisting more than 10% of total assemblage was considered as dominant species. We agree to show the composition table as supplementary materials in the revised version.

P10 line 31-33: More detailed explanation may be needed, like in the case of delta 15N in Fig. 6A and 6B. - Response: Trophic enrichment [Trophic enrichment (or fractionation factor) from basal food items based on the difference of each sample's delta 15N between higher trophic level to lower trophic level] was explained in the figure legend. Maybe our explanation makes the reviewer confused. We will explain more careful in Materials and methods in the revised version.

P10 line18: There are no results for Centropages in Fig. 4, Fig. 5 and Table 1. However, the authors showed the dietary compositions of Centropages as a major omnivorous copepod genera in Fig. 10D. Why? - Response: As the occurrence frequency of Centropages was low, so that the ïĄď15N estimated from linear mixing model was insignificant, thus we cannot obtain the isotope bi-plots, trophic level and trophic enrichment of this species. However, we obtained a significant value of ïĄď13C, and we use the average trophic enrichment of marine calanoids thus we could obtain the information of diet composition using SIAR package.

P11 line 29-32: The authors did not consider either cyclopoids or brackish water calanoids because those are not co-occurred with Labidocera, a surface water species.

However, I believe that Labidocera has a chance to contact other preys beside Acartia and Paracalanus, such as cyclopoids and brackish water calanoids. If the authors check the copepod community in summer, not only dominant species but other subdominant species (not shown in Table 1), there are many adult and copepodite copepod species that can be a potential prey for Labidocera. So, please add potential prey in Fig. 11. - Response: Agree to do so.

P11 line 32- P12 line 1-2: For Sinicalanus, potential prey including brackish water calanoid such as Pseudodiaptomus should be tested in Fig. 11. Also, I failed to understand that why Acartia was considered as prey for Sinocalanus in Fig. 11, considering Acartia was not dominant species in autumn in Table 1. - Response: Agree to add Pseudodiaptomus as potential prey for Sinocalanus. Acartia continuously occurred though they were not dominating in the autumn.

P14 Line23-25: I understand that calanoids (both marine and brackish water types) and cyclopoids had different delta 15N values according to Fig. 6B. However, the authors mentioned the mean value of the group was the same. Please check again. - Response: We may make confuse to the reviewer. In this sentence, we primarily discussed the similar trophic niche among the three major copepod group so that the referring figures should only contain Fig.4A and Fig.5. We will delete to refer Fig.6 in the parenthesis as Fig.6 was trophic enrichments on different sizes of plankton, which were estimated and averaged from all seasons and all stations.

P15 line 3: There is no result of the brackish water species, Pseudodiaptomus in Fig.4, but in Fig. 5. Why? - Response: We can add the result of Pseudodiaptomus in Fig.4.

P15 line 20-24: Corycaeus affinis was evaluated as omnivorous in this study, but as carnivorous in previous reports. What is a possible explanation for this difference? - Response: We believe that the feeding behavior of Corycaeus was not fully examined in literature. We have checked the existing literature more carefully about the feeding behavior of Corycaeus. However, we found that some reference suggests Corycaeus

are omnivores based on investigation of the contents of fecal pellets (Turner, 1986). There were some large-sized diatoms (Thalassiofhrix sp.) found in the pellets. Although Corycaeus can locate the prey visually, the can sometimes switch to filter-feeding and act as herbivores based on prey concentrations, according to investigation of the size and shape of the cutting edge of the mandibles (Giesecke & González, 2004)

P15 line 27-31: I believe decapod issue is not necessary for this study. Why did the authors include decapod results? - Response: We agree and will delete it in the revised version.

P15 line 31-33: There is no result of Euterpina as a genus of benthic harpacticoids in the results section, but only as harpacticoids. However, the authors mentioned Euterpina was detrivores in discussion and conclusion. In case of cyclopoids, the trophic level of cyclopoids and Corycaeus was presented separately in Fig. 4. Why? - Response: We directly measured the isotope values of the harpacticoid sample which primarily composed by Euterpina acifrons, while we didn't have the detail data of different harpacticoids. For Cyclopoids, we estimated the isotopic ratios of different genera and Corycaeus were significantly dominated.

P16 line 13-16: Even though Acartia dominated the marine calanoids in winter and summer, it is questionable to say that the bulk copepod assemblage with various species prefers large particles (microplankton; Fig. 7A and 7B). Likewise, Paracalanus also dominated the marine calanoid community in the more saline region in winter (Table 1), and Paracalanus prefers small particles (nanoplankton; Fig. 10). Paracalanus and other marine calanoids other than Acartia also may have contributed to the feeding selectivity of the bulk copepod assemblage differently. - Response: Yes, the feeding selectivity of the bulk copepod assemblage was a balance of ingestion among different groups. Our results showed a mean value of diet contribution of the bulk sample from all stations at a given season. When the bulk assemblage was shown preferring to feed on large-sized of POM, those species preferring large particle (e.g. Acartia) would play

a more important role in the assemblage in feeding prey. On the other hand, in the spring and autumn when the assemblage was primarily dominated by Corycaeus (our result suggested this genus was an omnivorous species and the size-selectivity was less pronounced), the assemblage overall didn't show an apparent size-selectivity.

P16 line 31: Corycaeus affinis dominated copepod community in spring and autumn, except for the river mouth. This result is inconsistent with previous reports in the same region; Corycaeus affinis was not a dominant species in spring and autumn (Kwon et al. 2001, Jang et al. 2004). I am very curious about the difference. My speculation is that horizontal net towing (0.5-1m depth) in the deeper region in this study may be responsible for potential bias of copepod composition. (Kwon KY, Lee PG, Park C, Moon CH, Park MO. 2001. Biomass and species composition of phytoplankton and zooplankton along the salinity gradients in the Seomjin River estuary. The Sea, J Korean Soc Oceanogr, 6: 9-102 Jang MC, Jang PG, Shin K, Park DW, Chang M. 2004. Seasonal variation of zooplankton community in Gwangyang Bay. Korean J Environ Biol, 22: 11-29) - Response: It is hard to give a correct speculation for this difference. However, I think annually variation due to ecosystem change is normal. This species is now quite common around the World Ocean and worth to study more carefully in the future. Nevertheless, we used the same sampling way between taxonomic data and isotopic data, as well as among different seasons. It wouldn't have any uncoupling of community composition and the trophic information of the assemblage.

P16 line 32: The authors concluded that Pseudodiaptomus was a detrivore, feeding on small phytoplankton cells. However, recent paper (Kayfetz and Kimmerer 2017) showed that P. forbesi in San Francisco Bay is rather omnivores feeding on various kinds of preys including centric diatom, pennate diatom, diatom (7-15_m), flagellates, flagellate (7-15_m), dinoflagellate and ciliate in the laboratory. (Kayfetz K, Kimmerer W. 2017. Abiotic and biotic controls on the copepod Pseudodiaptomus forbesi in the upper San Francisco Estuary. Mar Ecol Prog Ser, 581: 85-101) - Response: The reviewer may misunderstand our conclusion or our explanation may cause some confusing. We

found that Pseudodiaptomus were able to feed on plankton based on the mixing models and showed that Pseudodiaptomus preferred small-sized particle comparing the two major prey items (Fig.10 C). However, the $\delta$15N of Pseudodiaptomus estimated from the bulk sample was so low that we speculated that the detritus with low $\delta$15N may contribute to the balance the $\delta$15N of Pseudodiaptomus. Thus, we concluded that Pseudodiatomus were primarily an omnivorous species which preferred on small-sized particle by filter-feeding and was also strongly influenced by detritus.

P17 line 5-6: The authors mentioned that harpacticoids contributed to total copepod diet, preferring microplankton in winter (Fig. 7A), because harpacticoid preferred microplankton (Fig. 9D). However, harpacticoids are not a dominant group in winter (see Table 1). - Response: Although the contribution of harpacticoids to the total assemblage feeding may be weaker than the dominant species such as Acartia, the copepod feeding selectivity was a balance from all existing individuals including both dominating species and other species.

P17 line 9-13: The authors used the Bayesian mixing model to estimate the relative contribution of copepods to the carnivore diets, and the prey copepods which were not occurred with predatory copepods according to Table 1 were not considered in the model processing. However, this assumption or process may brings bias when evaluate the prey copepod contribution to predators in reality. The authors did not consider some copepod prey for Labidocera and Sinocalanus, but not Tortanus in Fig. 11. I guess that Labidocera who living on surface also may contact copepods other than Acartia and Paracalanus (for example, according to Table 1, in summer Labidocera rotunda co-occurred with Tortanus as well as Acartia spp.). Therefore, the brackish calanoids and cyclopoid also need to be included in potential prey for Labiocera. The same logic can be applied to Sinocalanus. Although Sinocalanus tellenus dominated in autumn with Paracalanus and Corycaeus, only Acartia was considered as prey for Sinocalanus, but not brackish water calanoid such as Pseudodiaptomus. Please consider all potential prey for Labidocera and Sinocalanus like in the case of Tortanus in

Fig. 11A. Also, it is not clear whether the dietary composition of the carnivorous genera in Fig. 10 was for a season or for the four seasons. Please specify appropriate season for each carnivorous copepods (e.g. all season or particular season) so that we can guess the potential prey for the carnivorous copepods. - Response: We will try to do so by considering all potential prey for Labidocera and Sinocalanus as suggested by the reviewer. By carefully check the taxonomic data set, we agree that the brackish calanoids Pseudodiaptomus should be included as a potential food source for Labiocera, as they co-occurred. However, when we observed Labidocera during the summer, we found that cyclopoid species didn't occur or may be in extremely low abundance. In such case we don't agree to consider cyclopoid species as potential food source for Labidocera. The dietary composition of the carnivorous genera in Fig. 10 was for all four seasons, as we used the all samples to estimate a mean isotope ratio for each genus.

P28 Fig.4: Please indicate which genera are the brackish calanoids or marine calanoids in Fig. 5(B) and/or Fig. 5. Also, please specify whether the result of decapods or harpacticoids is for spring and/or winter samples. - Response: I think the reviewer is saying the Fig.5 and Fig.6 (B). We have specified them in figure legend in the revised version.

P33 Fig.9: Please indicate appropriate season for each copepod group and decapods. - Response: Except decapods, all genera are averaged from all seasons, which we will indicate them in the figure legend in the revised version. And we will remove decapods, as suggested by the reviewer in one of the above comments.

Technical Corrections: P15 line 11: 'brackish stations in autumn and saline stations in winter' instead of 'brackish stations in winter and saline stations in autumn' - Response: Agree to revise.

P 15 line 24: 'Turner, 1984' instead of 'Turner, 1986' - Response: Agree to revise.

P16 line 20: 'Fig. 10B' instead of 'Fig. 9B' for Paracalanus - Response: Agree to

revise.

P17 line13: 'Sinocalanus preferred Paracalanus to Acartia and/or cyclopoids.' instead of 'Sinocalanus preferred cyclopoids to Acartia.' - Response: We have revised it according to revised mixing model analysis by including more potential food items, as suggested by the reviewer mentioned above.

---

## Author Comment (AC2) · 13 Dec 2017

General Comments: Chen et al report seasonal and spatial variations of copepods on 13C and 15N values in a temperate estuarine system. They present a nice description of these data and use a lot of mathematical analysis models (linear mixing models, Bayesian isotopic mixing models and generalized additive models) to deeply analyze the trophic structure of plankton. I am in favor of some salient results on averaged trophic position of different copepods and contribution of two size fractions of diets. These kinds of results are hard to be obtained by direct measure as copepod community is highly complex so that the individual samples are difficult to separate, which also

claimed by the authors. Although the size-selective feeding behaviors of copepods are not new in literature, the patterns shown in this manuscript are reasonable. More important, it still provides a powerful technique to treat such investigation data that can be followed by readers and provide insight biogeochemical information about the trophic interaction between copepods and primary producers. Therefore, this is potentially a very useful paper providing important information and methods for the biogeochemical study (i.e., food resources and trophic levels) in the complex coastal ecosystem, as well as the influence of the freshwater input in an estuary. The main shortfall of this manuscript is that it can only provide the trophic information of several major genera of copepods. Genera with low biomass or appearance frequency like Euchaeta, Calanus and Oithona, which are also popular in the world ocean cannot be treated by the same way. In addition, I would like to suggest more discussion about the uncertainty or disadvantages of these analysis models. And, reasons for some results on feeding pattern of some species are not discussed enough. For example, what are the mechanism of the feeding selectivity of the three carnivorous genera like Tortanus, Labidocera, and Sinocalanus? Finally, it will be more visual if the authors can provide a conceptual map about the planktonic food web from their conclusion, showing the relationship and the seasonal differences of the energy flow on this map as well? Overall, I recommend this manuscript for publication in Biogeosciences with minor revision. Some specific comments are indicated below. - Response: We appreciate for the reviewer's positive comments. We also admit the criticism of the shortfall of this paper. It is hard to estimate the isotope ratio of those species that contribute a very small fraction in total copepod biomass in Gwangyang Bay. However, we believe that the same way can be applied to Euchaeta, Calanus and Oithona when they dominate in community and have relatively high biomass. Such cases were commonly found in adjacent waters like East China Sea and South China Sea. As pointed out by other reviewers, we will also try to increase the discussion of the potential prey of the three carnivorous genera Tortanus, Labidocera, and Sinocalanus in the revised version.

Specific comments: 1. A reason or a reference to calculate trophic enrichment is
needed. - Response: We have added it to Page 7 Line 24. To remove confuse, we will also explain it more careful in figure legend.

2. L21 \_m 3. How about the errors or residual (eq. 4) for Linear regression models? - Response: We can add the errors of the model tests. 4. Y-axes in Figure 5 to 11 need plural number. - Response: We will revise accordingly.

5. Increase the resolution of Figure 3. - Response: We will input the separate pdf version for each figure.

---

## Author Comment (AC3) · 13 Dec 2017

General Comments: I find a problem in the way the authors estimated the weight differences between cyclopoids and calanoids randomly, as well as assuming that the weight of all calanoid genera was the same. In particular, because the authors have the taxonomic information already, I suggest they do a literature review and obtain the average weight values for each of the copepod genera/species used in the study, and apply these to the bulk regressions. I believe this is especially important as the authors are trying to extrapolate significantly more results than what they measured (i.e. genera-specific isotope values from a mixed community), that the approach be
as precise as possible. - Response: We thank the valuable recommendation of the reviewer. We will try to do a literature review and obtain the average weight values for the important genera used in our study.

In general I appreciate the effort to expand upon simple d13C and d15N bulk measurements for more detailed information on a community. However, in the case of copepods, if the authors do/did intend to investigate these relationships in detail, why not simply measure the values of individual genera? They state that too much material is required, but methodological advances these days mean that an individual Calanus female can indeed be analyzed (60ugC, 10ugN), as 5 ugN is typically the lower limit of standard bulk analyses (and low-N methods methods have been developed to go down to 1 ugN). Cyclopoids would require greater number, but following the authors assumptions of 0.1<x<1, that would be about 10 individuals. When certain problems arise, such as Paracalanus and Sinocalanus having lower d13C values than any measured prey, it would seem the authors acknowledge them, but then continue their analyses, e.g. calculate a TL (presumably based on prey that has been shown to not be consistent with their isotope values) in the same was as for the other genera. - Response: We agree to the reviewer's consideration. However, we missed to do so when the investigation was conducted. As stating in the ïĆšIntroductionïĆš part, it is too time-consuming to obtain enough weight for specific genus and isotope analysis for different subgroups requires great expertise in isolating species from highly complex mixtures. Besides, we try to simplify the sampling way so that some monitoring departments may follow.

One gets the sense that by plugging it into GAMs and regression models, the error sources and magnitudes are lost. I would like to see a quantitative test of the biases inherent in this Bayesian model, and how confident the authors can be that this approach is recovering the actual copepod diets. Given this approach and the number of assumptions that lie within, uncertainty relating to the model (as well as replication, independently) should be presented, discussed, and assessed explicitly with the other sources of uncertainty. This should be done with both the particle feeders and the

carnivorous species, and the effects of including or excluding different species types should also be assessed. - Response: The reliability of Bayesian mixing model is fully discussed in literature (Phillips and Koch, 2002; Phillips and Gregg, 2003; Moore and Semmens, 2008; Ward et al., 2010; Parnell et al., 2010, 2013). We are not good at extrapolating this model. However, we try to present more about the detail we used, e.g. the replication, trophic enrich factor, replications just like the reviewer suggest.

Finally, consistent with the point I discuss above, the authors mention a 'simple energy flow' in the abstract and discussion. But I wonder if this methodological approach allows for more complex flows. The actual isotopic values were not measured, but inferred from mass balance of dominant genera, and Bayesian approaches, and the violation of the underlying assumptions was not determined. How would a more complex picture emerge? In fact, the problem of Paracalanus and Sinocalanus having lower d13C values could hint at more complexity, yet it is assumed perhaps that this is due to unmeasured food sources and then ignored. I think if the authors address the issues posed above (and specifics below), the MS is suitable for publication. - Response: We admit that a complex picture cannot be fully understood from this paper. The estimated values only can provide a mean value and a standard error, thus they cannot explicit exactly the same situation of different seasons and no dynamics picture can be found. But the mechanism to estimate the isotopic ratio of a mixing sample, which is mixed by different species with different masses, is clear. Our results only suggest the potential trophic position of those examined genera. For Paracalanus and Sinocalanus, their low 13C was significantly estimated from the samples containing certain amount of them. We believe the data are true and correct. Although we fail to provide information of all potential food items for them, they show the role in interacting with plankton and other copepods. Frankly speaking, we didn't intend to give detail information of the biology of each genus, but aims to investigate their potential roles in regulating the abundance of the two size fractions of plankton.

Specific comments: Abstract. P1 – 10. The word 'trophism' is introduced yet does not

technically mean what the authors define it as (food resources and trophic levels), and is not used within the field's jargon as such either. I would prefer 'trophic structure' or 'trophic interactions', or 'trophic preferences'. - Response: Agree to change. We will use "trophic preference" in the revised version.

'Temperature-related' seasonal variations – The effect of temperature from season was not separated in this study, it should be simply 'seasonal variations'. - Response: Agree to change.

Introduction P2-0. "With broad feeding spectra and flexible feeding strategies, the bulk copepod assemblage is omnivorous depending on dominant species or group". Omnivo- rous or what? Consider changing to something like 'displays varying degrees of herbivory/ omnivory/carnivory, depending on dominant: : :' - Response: Agree to change.

P2-5. "In turn, TLs of a diverse: : :". I assume the authors here refer to the average trophic position of the assemblage, and thus should be 'TL' (singular). "Because cope-pods play a fundamental role in feeding on phytoplankton as primary consumers". Con-sider re-phrasing as 'Because copepods rely significantly on phytoplankton as prey', otherwise the expectation of this phrase is that the second half will refer to the topdown effect of copepods on phytoplankton, and not the bottom-up effect of phytoplankton on copepods. 'feeding on phytoplankton as primary consumers, so the seasonal and spatial'. Delete 'so'. - Response: Agree to change.

P2-15. "Therefore, the assessment of the trophic position (: : :) of copepods within a complex planktonic food web is critical in predicting the ecological relationships be-tween predator and prey". This phrase seems redundant, isn't the study about assess-ing these ecological relationships? I don't understand the prediction part. - Response: To avoid confuse, we will revise "in predicting" to "to understand".

P3. 0. "In contrast, the d15N values of primary producers increase from being nutrient sufficient (high fractionation) to nutrient-limiting (low fractionation) and are especially

high in anthropogenic wastewater nitrogen inputs". Would the later simple swamp the fractionation effect? The literature on ïAËŽd'15N of different nutrients in the ocean (nitrate, ammonia, urea) shows ranges that are much larger than fractionation factors, e.g. these vary by about 20‰ compared to 3.4‰ of fractionation. Can you comment on how much you expect the source to vary along the river gradient? - Response: The parentheses here indicate the consequences. For example, rich nutrients will cause high fractionation of primary producer. And they would continuously accumulate their $\delta$15N in the cells. We don't have the data on the variations of $\delta$15N of source, while we expect the $\delta$15N of source will vary from 0 to 13‰ based on the variances of POM.

Materials and methods P4-25. Could you mention the average volume filtered per tow, as the net was equipped with a flow meter? - Response: Sure, we can provide this.

P5-5. "water samples were transported to the laboratory as soon as possible". Please give a time estimate. - Response: 1-2 hours driving.

P6-5. The analytical precision of 0.2‰ and 0.3‰ for d13C and d15N, respectively, seems a bit high. Could you estimate what is the lowest change in TL that you can estimate based on this instrument error? - Response: We have re-checked the precision of the instrument during the period we measured the samples. The analytical precision should be 0.1 and 0.05 ‰ for $\delta$13C and $\delta$15N, respectively. Based on the equation of trophic level, the lowest change will be less than 0.02 TL.

P6.15. The weight difference between cyclopoids and calanoids was generated randomly. I don't understand why the information from the species identification was not used for this purpose. What is the error associated with this type of computation? I would really suggest the authors do a literature search of the mean weights of the difference species and genera enumerated in their samples, and use this information to estimate both cyclopoid/calanoid weights, and the weights of the different calanoid genera. If the composition has already been estimated, it makes no sense to make these assumptions that only introduce greater error into an already indirect way of estimating species stable isotope composition. - Response: As mentioned above, we will do a literature search.

P7-20. 'fractionation factors used in the model estimation were calculated from TLs'. I don't understand this statement, it sounds like 3‰ and 0.5 ‰ were assumed (logically) and not calculated. Please clarify. - Response: The sentence of "fractionation factors used in the model estimation were calculated from TLs' means that the fractionation factor between two trophic levels, based on the difference of each $\delta15N$ value. Or it can also be understood by this way, it is multiplying the difference of two trophic levels (calculated from different $\delta15N$ value) with 3‰ and 0.5 ‰ for $\delta15N$ and $\delta13C$, respectively.

Results The authors discuss their seasonal results in the context of 'temperature'. I would prefer to see this discussed as 'seasonal', since temperature variability within a season was not tested and hence the driver of the observed effects cannot be unequivocally stated to be temperature. Rather, they are probably a combined effect of the changes that co-occur with each season and should be stated as such. - Response: We do test the partial effect of temperature on variation of plankton isotope based on GAM results (Fig.3). However, we agree to avoid too much emphasis on temperature but on season.

P8-10. 'Despite insignificant spatial variability, higher Chl a concentrations generally occurred in the middle of the bay'. This is not obvious from the values in the table. Please explain in more detail or remove. - Response: Agree to revise.

P8-25. Please give a mean value for copepod d13C as done for the groups above (nanoplankton and microplankton) - Response: Agree to revise.

P9-0. "Overall, seasonal succession of winter-spring, spring-summer, and summer autumn were apparent for all plankton groups". Not clear what this means. There appears to be significant overlap in values for the nanoplankton, and no clear increasing progression from winter to autumn, as increases/decreases seem to interchange. -

Response: Yes, seasonal pattern for Nanoplankton was not clear. We decide to delete this sentence.

P9-5. It isn't clear to me how the coefficients of variation are calculated. The range of d15N values encountered is less than that for d13C, although the spatial progressions are less monotonic. Please clarify in the methods how this is calculated. - Response: We calculated the coefficients of variation by dividing the standard error with mean value. We will clarify this in the M&M.

P9-10. The result for the microplankton is inconsistent with the figure. In the figure, the highest value for d15N is 10‰ at the bay in spring. There is no 16.2 value. - Response: We may have a typo. It should be 10.2 at station 9. Thanks for the reviewer's careful check.

P9-15. "Copepod d15N : : : being much more consistent with the pattern of microplankton than that of nanoplankton". This seems true for the summer ïAËŻd'15N values, and quite the opposite for the winter values. Regardless, there is such high variability that it is hard to tease out any clear pattern of spatial/seasonal co-variability. - Response: Yes, it is true. We will remove this unsuitable sentence.

P9-20. The GAM result is very interesting. Perhaps it reflects the food-web processes that affect d15N disproportionally and were not included in the GAM? Response: Here the deviances explained by GAM suggest those factors combined to influence the dependent factor. For $\delta$15N, the deviance explained was relatively lower suggesting that other factors which were not included would contribute another 23% of the deviance of $\delta$15N. But we don't know what are them. The understanding of the reviewer is right.

P10-20. It is not clear to me how the trophic levels of brackish copepods can be calculated, when their 13C values do not support the sampled nanoplankton and/or microplankton as their food source. I also don't understand how later in figure 6 they show up enriched, but in figure 4 they are depleted with respect to this food source. The differences between these two figures should be stated clearly as they show different results. - Response: The brackish copepods in this study were defined by empirical taxonomy, including Pseudodiaptmus and Sinocalanus. The trophic levels were calculated by the formula shown in M&M and figure legend. We admit that there may be some confuse, while we don't think the two figures show different results. Firstly, the results of brackish copepods were averaged from Pseudodiaptmus and Sinocalanus, while our result showed more insight information that they were different. Secondly, as mentioned above, we didn't intend to give detail information of the biology of each genus, but aims to investigate their potential roles in regulating the abundance of the two size fractions of plankton. Thus, we were unable to provide detail information of all potential diet sources of them, whereas they still have enrichment factor on lower trophic levels such as nanoplankton and microplanton.

P10-25. "The enrichment values for nanoplankton feeding on marine and brackish water calanoids: : :". This phrase says that nanoplankton are feeding on copepods. That's not right, it should say something like 'enrichment values for marine and brackish water: : : feeding on nanoplankton'. Response: Sorry for a mistake here. We will revise as suggested.

P11-5. I disagree with the statement (based on the figure) that "the proportions of the two size fractions of POM averaged from all four seasons contributing to copepod diets at different stations were also distinctly different except for station 8 (Fig. 8)". It seems that the error bars overlap at station 1 (hence not different), and stations 6 and 7. I might be missing something but then it should be clarified. - Response: Yes, we found that error bars were indeed overlapping. We will revise this conclusion.

P11-10. Does 'spring data available' mean 'only spring shown'? - Response: Yes, we obtained enough amounts of decapods for isotopic analysis only at the spring. However, as suggested by another reviewer, we decide to delete the part of decapods as it was not related to the topic of this study.

The authors discuss size-selective feeding of calanoids in the context of 'filtering efficiency', yet they are not true filter feeders, they are suspension feeders that trap and handle particles (Paffenhofer et al, 1982, Mar Bio 67:2), which has different implications for particle handling. This is an important distinction that should be observed throughout the MS. - Response: OK, we will carefully check the whole MS and change to "feeding efficiency"

Discussion P13-0. It seems to me that the sewage explanation deserves a bit more attention. If the authors can't rule it out it means that this could contribute substantially and swamp the other subtle processes discussed in the 15N-enriched ammonia section. - Response: Yes, we also believe that sewage was important for 15N accumulating. However, we didn't have direct data to support our speculation. Thus we are going to change the sentence (P13 Line 5-6) to "The input of sewage-derived 15N-enriched ammonia 5 (domestic sewage and livestock waste) could contribute substantially and swamp the other subtle processes to increase $\delta$15N values of nanoplankton."

P13-5. "Furthermore, the fractionation effect of phytoplankton will be reduced when phytoplankton became nitrogen-limited and take up nitrogen with little fractionation". I am unsure that this effect could be significant in a coastal areas such as this one. Moreover, if phytoplankton reduce their fractionation, it would mean that their 15N will tend to be higher (as they choose the lighter 14N), and thus doesn't explain this decreasing trend. - Response: Yes, we agree that nutrient-limiting is not frequently happened in coastal area. Nevertheless, substantial reduction of nutrients from different seasons or from different stations and the mis-match of high phytoplankton and low nutrients were normal. When phytoplankton reduce fractionation, they will select more ligther 14N in cells thus they will show a reducing ratio of 15N in cells (Cifuentes et al., 1988; Fogel and Cifuentes, 1993; Granger et al., 2004). To remove such confuse, we will revise this sentence to ""Furthermore, the fractionation effect of phytoplankton will be reduced when nutrients substantially decreased and phytoplankton would take up nitrogen with little fractionation and stored relatively light of nitrogen isotope."

P13-10. I would like to see table with the GAM results. It would be nice to have these

presented first in the results, and later discussed. It would also be interesting to see the different variables tested and the ones found to be significant within this table. - Response: Agree to revise. We will show the table in Results. And as suggested by another reviewer, we will move the GAM figures (Fig. 3) to supplementary materials.

P13-20. But see Gutierrez-Rodriguez et al (2014, L&O, vol:59, i5) on negligible trophic enrichment of heterotrophic protists. - Response: Thank the reviewer's reference. We agree the negligible trophic enrichment of heterotrophic protists, thus we will remove such speculation in discussion.

P14-0. "Because of different feeding behaviors and fractionation effects of copepods, the variability of trophic positions of copepod assemblage depends on the overall composition of species and is determined by dominant species." Change to ": : :the variability of the average community trophic position depends on the overall composition of species and is determined by the dominant species." - Response: Agree to revise.

I am somewhat confused about the discussion of trophic levels of the copepods Paracalanus and Sinocalanus. The authors state that their ïAËŽd'13C values are lower than all measured food sources, which would imply that their food source has not been adequately measured. How then are these organisms included in the trophic level (TL) component of the paper? A bit of clarification on this topic would really help the reader. Response: Agree to do so. The trophic level in this study was defined as trophic position relative to Nanoplankton, which was considered as the trophic baseline.

P17-10. This paragraph explaining the Bayesian mixing model methods/results should be moved to the results section. - Response: Agree to revise.

---

## Author Comment (AC4) · 13 Dec 2017

General Comments This manuscript provides results from seasonal and spatial variation in the stable isotopes 13C and 15N of POM and copepods along a salinity gradient in Gwangyang Bay, off the southern coast of Korea. The authors combined this information with linear mixing models, Bayesian isotopic mixing models and generalized additive models to derive a statement on food selectivity and trophic level of copepods. In general, this manuscript is very well structured and provides valuable information on the flow of matter through the food web. Still, some concerns have to be clarified before publication. - Response: We appreciated the positive comments of the reviewer

and will follow the suggestion to improve the manuscript.

Specific comments. Introduction 1. Page 3, line 7: Please give more information here on the usage of different N sources and enrichment factors. - Response: Agree. Accordingly, we will explain more information here based on literature.

2. Page 3, line 19: "highly mixed species"- Please clarify, mixed with what? - Response: Here the "highly mixed species" means the assemblage contained too many different species and those species had similar size. So such species were hard to be sorted out from the assemblage based on current microscopic technique. To remove confuse, we will revise it to "high diversity of the assemblage and ..." .

3. Page 3, line 21: Instrument sensitivity has increased and compound specific analysis (CSI) of stable isotopes in amino acids make it possible to track diets of mesozooplankton and determine their trophic position. - Response: Yes, of course. We admit that highly developed instrument can do so. But for doing so, researchers still need taxonomic expertise to sort out the species from a complex mixture to prepare the sub-sample. It requires a lot of lab processing works.

4. Page 3, line 21: Please give some reason why this site was chosen. - Response: The stations were chosen based on salinity regime and different geographic characteristics, e.g. stations 1~3 are river sites with extremely low salinity, stations 4~6 are in the central bay with moderate salinity, while stations 7~9 are in the channel towards to the open ocean with relatively high salinity.

Material and Methods 5. General: why did the authors not use literature data on average weight values for each of the species investigated instead of assigning the weight to each group? - Response: In the revised version, we will search the literature data just like suggestion of this comment and also suggested by other reviewers.

6. General: How where copepodite stages treated regarding abundance and body mass? - Response: They were averaged to adults.

[Figure]

7. Page 4, line 15: Change to "increasing". - Response: Agree and we will revise accordingly.

8. Page 4, line 16: Specify "in the middle of Gwangyang Bay. - Response: Agree. We will revise to ïĆšin the middle part of the Gwangyang BayïĆš.

9. Page 4: Please add information on when sampling took place- day or night? - Response: We all sample at the day time. We will add such explanation in M&M.

10. Page 5, line 11: "pico- and nano- sized phytoplankton". Doesn' t sampling with a mesh also include nanozooplankton like heterotorphic and mixotrophic flagellates- so it does not only comprise phytoplankton?! - Response: Here the plankton less than 20 micron but larger than GF/F (0.78 micron) were defined as nanoplankton. Thus they contains both phytoplankton and heterotrophs.

11. Page 6, line 29: something is missing at the end of the sentence- "illustrated in figures?". - Response: The figures here do not mean citations. We try to explain that the mean and standard deviations were illustrated by forms of figures. To remove confuse, we can delete this sentence in the revised version.

Results and Discussion 12. There are too many figures. Some might be moved to the supplemental section, e.g. Fig. 3, 7,8,11 - Response: We agree to do so. We plan to move Fig.3, 7 and 8 to supplementary materials, but no Fig.11. We believe that Fig.11 is relatively important for readers and other reviewer want to know more about the information of the feeding of carnivorous species.

13. Page 12, line 16: What is a "heavy carbon pool", give an example? - Response: The phrase is located at "Page 12, line 24". "Heavy carbon pool" here means the dissolved inorganic carbon pool in which the carbon was primarily composed by light carbon (12C).

14. Page 12, line 31: Wording! Please revise "much reduced". - Response: We will change it to "low".

15. Page 13, line 15: "with low fractionation effects"- give example. - Response: We will search more suitable examples from literature for this comment.

Conclusion 16. Please provide a simplified figure of the energy flow for the different seasons. - Response: Based on revised estimation, we try to provide such simplified figures.

---

## Author Response (AR1)

**Journal: BG**

**Title: Variability in copepod trophic levels and in feeding selectivity based on stable isotope analysis in Gwangyang Bay off the southern coast of Korea**

**Author(s): Mianrun Chen et al.**

**MS No.: bg-2017-364**

**MS Type: Research article**

**Rebuttals and responses to reviews**

We appreciate four anonymous reviewers for the constructive comments and extremely useful reviews on the manuscript. The texts referred to by the reviewer are indicated, in the responses, by page and line numbers of the revised version. We have tried to revise carefully in line with the suggestions made by the reviewers as follows:

Reply to Referee #1

**General Comments:**

The authors used stable isotope analysis to solve copepod trophism (i.e. food resources and trophic level), which is important to understand the biogeochemistry in estuarine system. The findings on copepod trophism in the manuscript (MS) will contribute for understanding pelagic food webs in the system. These are valuable and positive points in the MS.

Nevertheless, I found many doubtful points throughout the MS. The authors simplified the dynamics of copepod community by considering the most dominant copepod species only, and then applied this simplified copepod assemblage to the stable isotope analysis. As a result, trophism of some copepod group, especially carnivorous copepods are still questionable.

The current method (e.g. Bayesian mixing model) and the assumption (e.g. the body mass of different genera among calanoids are the same) applied to stable isotope analysis may have some limitation to evaluate the real trophism of the copepods in the

field, even though most results of copepod trophism in the MS were similar to previous reports.

Therefore, I would like to recommend the authors to include an additional explanation on a potential limitation which may occur when you apply the current method and assumption to copepod community, in the revised MS.

Response: We thank the reviewer for his/her careful checking of the manuscript and we agree with his/her general assessment of our work and are happy that he found our valuable and positive points in the MS. We agree that the current method has some limitations to study the entire complex planktonic structure. However, the reviewer may misunderstand our data set as we were not just considering the most dominant species only whereas we considered all appearing copepod genera in statistics when we interpreted the dynamics of the entire copepod community or specific taxonomic groups like calanoids and cyclopoids. We believed that copepod genus can be grouped based on similar feeding behaviors and thus the food web structure can be simplified.

**Specific Comments:**

P6, line 20-22: The author's assumption is questionable. Calanoids consist of many genera or species with various sizes. Even though some large calanoids are not dominant in the sample in terms of abundance, some large calanoids (e.g. *Calanus*) can have important role in terms of biomass o in the revised msr volume. So, the author's assumption may not apply to a mixed copepod community with existence of both small and large copepods.

Response: We admit that this assumption was a little bit inaccurate. Unfortunately, we didn't record the body length of our taxonomic data. As suggested by another reviewer, we will re-calculate their biomasses based on the empirical formula of biomass and body length of different copepod genus and use the ratio of body length among different genera. (see Supplementary Table S1 and relating results and discussion in the revised ms)

In relation to this issue, how did the authors treat copepodite stages of the copepods occurred in this study to calculate their abundance or body mass? There is no explanation in the materials and methods.

Response: Copepodites were counted but grouped together with adults. (see P5, Line 9 in the revised ms)

P8, line 13-19 and Table 1: There is no criterion for dominant species in Table 1. The authors listed only the most abundant copepod species by station and season. I think more than one copepod species would have contributed to copepod community in the field. Please specify a criterion and also show other copepod species if possible (in fact, the information on copepod species composition is poor and not informative in this study).

Response: Species with abundance consisting more than 10% of total assemblage was considered as dominant species. We agree to show the composition table as supplementary materials in the revised version. (see Supplementary Table S2 and P8, Lines 13–14 in the revised ms)

P10 line 31-33: More detailed explanation may be needed, like in the case of delta 15N in Fig. 6A and 6B.

Response: We have added more explanation on it. (see P11, Lines 14–18 in the revised ms)

P10 line18: There are no results for *Centropages* in Fig. 4, Fig. 5 and Table 1. However, the authors showed the dietary compositions of *Centropages* as a major omnivorous copepod genera in Fig. 10D. Why?

Response: Based on new method, we now have obtained the result of *Centropages*. (see Figs. 3–6 and relating result and discussion in the revised ms)

P11 line 29-32: The authors did not consider either cyclopoids or brackish water

calanoids because those are not co-occurred with Labidocera, a surface water species. However, I believe that Labidocera has a chance to contact other preys beside Acartia and Paracalanus, such as cyclopoids and brackish water calanoids. If the authors check the copepod community in summer, not only dominant species but other sub dominant species (not shown in Table 1), there are many adult and copepodite copepod species that can be a potential prey for Labidocera. So, please add potential prey in Fig. 11.

Response: Agree to do so. (see Fig.8A and relating results and discussion in the revised ms)

P11 line 32- P12 line 1-2: For *Sinicalanus*, potential prey including brackish water calanoid such as *Pseudodiaptomus* should be tested in Fig. 11. Also, I failed to understand that why Acartia was considered as prey for *Sinocalanus* in Fig. 11, considering *Acartia* was not dominant species in autumn in Table 1.

Response: Agree to add *Pseudodiaptomus* as potential prey for *Sinocalanus*. *Acartia* continuously occurred though they were not dominating in the autumn. (see Fig. 8B and relating results and discussion in the revised ms)

P14 Line23-25: I understand that calanoids (both marine and brackish water types) and cyclopoids had different delta 15N values according to Fig. 6B. However, the authors mentioned the mean value of the group was the same. Please check again.

Response: We may make confuse to the reviewer. In this sentence, we primarily discussed the similar trophic niche among the three major copepod groups so that the referring figures should only contain Fig. 4A and Fig. 5. We will delete to refer Fig.6 in the parenthesis as Fig.6 was trophic enrichments on different sizes of plankton, which were estimated and averaged from all seasons and all stations. Based on revised modeling test, they were indeed different trophic niche. We have revised accordingly. (see P15, Lines 9–14 in the revised ms)

P15 line 3: There is no result of the brackish water species, *Pseudodiaptomus* in Fig.4,

but in Fig. 5. Why?

Response: We have added the result of *Pseudodiaptomus*in Fig.4. (see Fig. 3B and P10, Lines 13–24 in the revised ms in the revised ms)

P15 line 20-24: Corycaeus affinis was evaluated as omnivorous in this study, but as carnivorous in previous reports. What is a possible explanation for this difference?

Response: Based on the revised version of model test, *Corycaeus affinis* was indeed a carnivorous species. We have revised accordingly. (see Fig. 3B, Fig. 8C, P16, Lines 2–6)

P15 line 27-31: I believe decapod issue is not necessary for this study. Why did the authors include decapod results?

Response: We agree and have deleted it in the revised version. (see Figs. 3–7 in the revised ms).

P15 line 31-33: There is no result of Euterpina as a genus of benthic harpacticoids in the results section, but only as harpacticoids. However, the authors mentioned Euterpina was detrivores in discussion and conclusion. In case of cyclopoids, the trophic level of cyclopoids and Corycaeus was presented separately in Fig. 4. Why?

Response: We directly measured the isotope values of the harpacticoid sample which primarily composed by *Euterpina acifrons*, while we didn't have the detail data of different harpacticoids. For Cyclopoids, we estimated the isotopic ratios of different genera and *Corycaeus* were significantly dominated. (see Figs. 3 and P16, Lines 5–9 in the revised ms).

P16 line 13-16: Even though Acartia dominated the marine calanoids in winter and summer, it is questionable to say that the bulk copepod assemblage with various species prefers large particles (microplankton; Fig. 7A and 7B). Likewise, Paracalanus also dominated the marine calanoid community in the more saline region in winter (Table1),

and Paracalanus prefers small particles (nanoplankton; Fig. 10). Paracalanus and other marine calanoids other than Acartia also may have contributed to the feeding selectivity of the bulk copepod assemblage differently.

Response: Yes, the feeding selectivity of the bulk copepod assemblage was a balance of ingestion among different groups. Our results showed a mean value of diet contribution of the bulk sample from all stations at a given season. When the bulk assemblage was shown preferring to feed on large-sized of POM, those species preferring large particle (e.g. *Acartia*) would play a more important role in the assemblage in feeding prey. On the other hand, in the spring and autumn when the assemblage was primarily dominated by carnivorous species (*Corycaeus*, *Tortanus*) and dominated by both *Paracalanus* (preferring small-sized particle) and *Acartia* (preferring large-sized particle), the assemblage overall didn't show an apparent size-selectivity. (see P16 Lines 16–29 and P17, Lines 3–17 in the revised ms)

P16 line 31: Corycaeus affinis dominated copepod community in spring and autumn, except for the river mouth. This result is inconsistent with previous reports in the same region; Corycaeus affinis was not a dominant species in spring and autumn (Kwon et al. 2001, Jang et al. 2004). I am very curious about the difference. My speculation is that horizontal net towing (0.5-1m depth) in the deeper region in this study may be responsible for potential bias of copepod composition. (Kwon KY, Lee PG, Park C, Moon CH, Park MO. 2001. Biomass and species composition of phytoplankton and zooplankton along the salinity gradients in the Seomjin River estuary. The Sea, J Korean Soc Oceanogr, 6: 9-102 Jang MC, Jang PG, Shin K, Park DW, Chang M. 2004. Seasonal variation of zooplankton community in Gwangyang Bay. Korean J Environ Biol, 22: 11-29)

Response: It is hard to give a correct speculation for this difference. However, I think annually variation due to ecosystem change is normal. This species is now quite common around the World Ocean and worth to study more carefully in the future. Nevertheless, we used the same sampling way between taxonomic data and isotopic data, as well as among different seasons. It wouldn't have any uncoupling of

community composition and the trophic information of the assemblage. (see P18 Lines 12–17 in the revised ms)

P16 line 32: The authors concluded that Pseudodiaptomus was a detrivore, feeding on small phytoplankton cells. However, recent paper (Kayfetz and Kimmerer 2017)showed that P. forbesi in San Francisco Bay is rather omnivores feeding on various kinds of preys including centric diatom, pennate diatom, diatom (7-15_m), flagellates, flagellate (7-15_m), dinoflagellate and ciliate in the laboratory. (Kayfetz K, Kimmerer W. 2017. Abiotic and biotic controls on the copepod Pseudodiaptomus forbesi in the upper San Francisco Estuary. Mar Ecol Prog Ser, 581: 85-101)

Response: The reviewer may misunderstand our conclusion or our explanation may cause some confusing. We found that *Pseudodiaptomus* were able to feed on plankton based on the mixing models and showed that *Pseudodiaptomus* preferred small-sized particle comparing the two major prey items (Fig.10 C). However, the $\delta^{15}N$ of *Pseudodiaptomus* estimated from the bulk sample was so low that we speculated that the detritus with low$\delta^{15}N$ may contribute to the balance the$\delta^{15}N$ of *Pseudodiaptomus*. Thus, we concluded that *Pseudodiaptomus* were primarily an omnivorous species which preferred on small-sized particle by filter-feeding and was also strongly influenced by detritus. (see P 17, Lines 26–33 and P18, Lines 1–8 in the revised ms)

P17 line 5-6: The authors mentioned that harpacticoids contributed to total copepod diet, preferring microplankton in winter (Fig. 7A), because harpacticoid preferred microplankton (Fig. 9D). However, harpacticoids are not a dominant group in winter (see Table 1).

Response: Although the contribution of harpacticoids to the total assemblage feeding may be weaker than the dominant species *Acartia*, the copepod feeding selectivity was a balance from all existing individuals including both dominating species and other species. (see P18 Lines 15–18 in the revised ms)

P17 line 9-13: The authors used the Bayesian mixing model to estimate the relative contribution of copepods to the carnivore diets, and the prey copepods which were not occurred with predatory copepods according to Table 1 were not considered in the model processing. However, this assumption or process may brings bias when evaluate the prey copepod contribution to predators in reality. The authors did not consider some copepod prey for Labidocera and Sinocalanus, but not Tortanus in Fig. 11. I guess that Labidocera who living on surface also may contact copepods other than Acartia and Paracalanus (for example, according to Table 1, in summer Labidocera rotunda co-occurred with Tortanus as well as Acartia spp.). Therefore, the brackish calanoids and cyclopoid also need to be included in potential prey for Labidocera. The same logic can be applied to Sinocalanus. Although Sinocalanus tellenus dominated in autumn with Paracalanus and Corycaeus, only Acartia was considered as prey for Sinocalanus, but not brackish water calanoid such as Pseudodiaptomus. Please consider all potential prey for Labidocera and Sinocalanus like in the case of Tortanus in Fig. 11A.Also, it is not clear whether the dietary composition of the carnivorous genera in Fig. 10was for a season or for the four seasons. Please specify appropriate season for each carnivorous copepods (e.g. all season or particular season) so that we can guess the potential prey for the carnivorous copepods.

Response: We will try to do so by considering all potential prey for *Labidocera* and *Sinocalanus* as suggested by the reviewer. By carefully check the taxonomic dataset, we agree that the brackish calanoids *Pseudodiaptomus* should be included as a potential food source for *Labiocera*, as they co-occurred. However, in our taxonomic data set, when we observed *Labidocera* during the summer, we found that cyclopoid species and didn't occur or may be in extremely low abundance. In such case we don't agree to consider cyclopoid species as potential food source for *Labidocera*. The dietary composition of the carnivorous genera in Fig. 10 was for all four seasons, as we used the all samples to estimate a mean isotope ratio for each genus. (see Fig. 8A, B, and P17, Lines 11–33, P18, Lines 1–4 and 8–11)

P28 Fig.4: Please indicate which genera are the brackish calanoids or marine calanoids in Fig. 5(B) and/or Fig. 5. Also, please specify whether the result of decapods or harpacticoids is for spring and/or winter samples.

Response: I think the reviewer is saying the Fig.5 and Fig.6 (B). We have specified them in figure legend in the revised version. (see Fig. 3 and Fig. 4B, and P6, Lines 16–22 in the revised ms)

P33 Fig.9: Please indicate appropriate season for each copepod group and decapods.

Response: Except decapods, all genera are averaged from all seasons, which we indicate them in the figure legend in the revised version. And we have removed decapods, as suggested by the reviewer in one of the above comments. (see Figs. 3–6 in the revised ms)

Technical Corrections:

P15 line 11: 'brackish stations in autumn and saline stations in winter' instead of 'brackish stations in winter and saline stations in autumn'

Response: Agree to revise. (see P15, Line 22 in the revised ms)

P 15 line 24: 'Turner, 1984' instead of 'Turner, 1986'

Response: Agree to revise. (see P16, Line 6 in the revised ms)

P16 line 20: 'Fig. 10B' instead of 'Fig. 9B' for Paracalanus

Response: Agree to revise. Now it is Fig. 7E. (see P17, Line 6 in the revised ms)

P17 line13: 'Sinocalanus preferred Paracalanus to Acartia and/or cyclopoids.' Instead of 'Sinocalanus preferred cyclopoids to Acartia.'

Response: We have revised it according to revised mixing model analysis by including more potential food items, as suggested by the reviewer mentioned above. (see P18, Line 5–8 in the revised ms)

Reply to Referee #2

**General Comments:**

Chen et al report seasonal and spatial variations of copepods on 13C and 15N values in a temperate estuarine system. They present a nice descriptionof these data and use a lot of mathematical analysis models (linear mixing models,Bayesian isotopic mixing models and generalized additive models) to deeply analyzethe trophic structure of plankton. I am in favor of some salient results on averagedtrophic position of different copepods and contribution of two size fractions of diets.These kinds of results are hard to be obtained by direct measure as copepod communityis highly complex so that the individual samples are difficult to separate, which alsoclaimed by the authors. Although the size-selective feeding behaviors of copepods arenot new in literature, the patterns shown in this manuscript are reasonable. More important,it still provides a powerful technique to treat such investigation data that can befollowed by readers and provide insight biogeochemical information about the trophicinteraction between copepods and primary producers. Therefore, this is potentially avery useful paper providing important information and methods for the biogeochemicalstudy (i.e., food resources and trophic levels) in the complex coastal ecosystem, aswell as the influence of the freshwater input in an estuary.The main shortfall of this manuscript is that it can only provide the trophic information ofseveral major genera of copepods. Genera with low biomass or appearance frequencylike Euchaeta, Calanus and Oithona, which are also popular in the world ocean cannotbe treated by the same way. In addition, I would like to suggest more discussion aboutthe uncertainty or disadvantages of these analysis models. And, reasons for someresults on feeding pattern of some species are not discussed enough. For example,what are the mechanism of the feeding selectivity of the three carnivorous genera like Tortanus,Labidocera, and Sinocalanus? Finally, it will be more visual if the authorscan provide a conceptual map about the planktonic food web from their conclusion,showing the relationship and the seasonal differences of the energy flow on this mapas well? Overall, I recommend this manuscript for publication

in Biogeosciences with minor revision. Some specific comments are indicated below.

Response: We appreciate for the reviewer's positive comments. We also admit the criticism of the shortfall of this paper. It is hard to estimate the isotope ratio of those species that contribute a very small fraction in total copepod biomass in GwangyangBay. However, we believe that the same way can be applied to *Euchaeta*, *Calanus* and *Oithona* when they dominate in community and have relatively high biomass. Such cases were commonly found in adjacent waters like East China Sea and South China Sea. As pointed out by other reviewers, we have also tried to increase the discussion of the potential prey of the three carnivorous genera *Tortanus*,*Labidocera*, and *Sinocalanus* in the revised version. (see Fig. 8, P17, Lines 26–33 and P18, Lines 1–11 in the revised ms)

**Specific comments**:

1. A reason or a reference to calculate trophic enrichment is needed.

Response: We have added it to Page 7 Line 24. To remove confuse, we have also explained it more careful in figure legend. (see Fig. 5 and P7, Lines 21–23 in the revised ms)

2. L21 _m

3. How about the errors or residual (eq. 4) for Linear regression models?

Response: We have addd the errors of the model tests. (see P10, Lines 4–5 and Lines 13–14 in the revised ms)

4. Y-axes in Figure 5 to 11 need plural number.

Response: We have revised accordingly. (see Figs. 4–8 in the revised ms)

5. Increase the resolution of Figure 3.

Response: We have done in the separate pdf version for each figure. (see all figures in the revised ms)

**Reply to Referee #3**

**General Comments:**

> I find a problem in the way the authors estimated the weight differences between cyclopoids and calanoids randomly, as well as assuming that the weight of all calanoid genera was the same. In particular, because the authors have the taxonomic information already, I suggest they do a literature review and obtain the average weight values for each of the copepod genera/species used in the study, and apply these to the bulk regressions. I believe this is especially important as the authors are trying to extrapolate significantly more results than what they measured (i.e. genera-specific isotope values from a mixed community), that the approach be as precise as possible.

Response: We thank the valuable recommendation of the reviewer. We have tried to do a literature review and obtain the average weight values for the important genera used in our study. (see Supplementary Table S1 in the revised ms)

> In general I appreciate the effort to expand upon simple d13C and d15N bulk measurements for more detailed information on a community. However, in the case of copepods, if the authors do/did intend to investigate these relationships in detail, why not simply measure the values of individual genera? They state that too much material is required, but methodological advances these days mean that an individual Calanus female can indeed be analyzed (60ugC, 10ugN), as 5 ugN is typically the lower limit of standard bulk analyses (and low-N methods methods have been developed to go down to 1 ugN). Cyclopoids would require greater number, but following the authors assumptions of 0.1<x<1, that would be about 10 individuals. When certain problems arise, such as Paracalanus and Sinocalanus having lower d13C values than any measured prey, it would seem the authors acknowledge them, but then continue their analyses, e.g. calculate a TL (presumably based on prey that has been shown to not be consistent with their isotope values) in the same was as for the other genera.

Response: We agree to the reviewer's consideration. However, we missed to do so when the investigation was conducted. As state in the "Introduction" part, it is too time-consuming to obtain enough weight for specific genus and isotope analysis for different subgroups requires great expertise in isolating species from highly complex mixtures. Besides, we have tried to simplify the sampling way so that some monitoring departments may follow. (see P6, Lines 13–25 in the revised ms)

> One gets the sense that by plugging it into GAMs and regression models, the error sources and magnitudes are lost. I would like to see a quantitative test of the biases inherent in this Bayesian model, and how confident the authors can be that this approach is recovering the actual copepod diets. Given this approach and the number of assumptions that lie within, uncertainty relating to the model (as well as replication, independently) should be presented, discussed, and assessed explicitly with the other sources of uncertainty. This should be done with both the particle feeders and the carnivorous species, and the effects of including or excluding different species types should also be assessed.

Response: The reliability of Bayesian mixing model is fully discussed in literature (Phillips and Koch, 2002; Phillips and Gregg, 2003; Moore and Semmens, 2008; Ward et al., 2010; Parnell et al., 2010, 2013). We are not good at extrapolating this model. However, we have tried to present more about the details we used, e.g. the replication, trophic enrich factor, sources just like the reviewer suggest. (see P7, Lines 11–27 in the revised ms)

> Finally, consistent with the point I discuss above, the authors mention a 'simple energy flow' in the abstract and discussion. But I wonder if this methodological approach allows for more complex flows. The actual isotopic values were not measured, but inferred from mass balance of dominant genera, and Bayesian approaches, and the violation of the underlying assumptions was not determined. How would a more complex picture emerge? In fact, the problem of Paracalanus and Sinocalanus having

lower d13C values could hint at more complexity, yet it is assumed perhaps that this is due to unmeasured food sources and then ignored.

I think if the authors address the issues posed above (and specifics below), the MS is suitable for publication.

Response: We admit that a complex picture cannot be fully understood from this paper. The estimated values only can provide a mean value and a standard error, thus they cannot explicit exactly the same situation of different seasons and no dynamics picture can be found. But the mechanism to estimate the isotopic ratio of a mixing sample, which is mixed by different species with different masses, is clear. Our results only suggest the potential trophic position of those examined genera. For *Paracalanus* and *Sinocalanus*, their low 13C was significantly estimated from the samples containing certain amount of them. We believe the data are true and correct. Although we fail to provide information of all potential food items for them, they show the role in interacting with plankton and other copepods. Frankly speaking, we didn't intend to give detail information of the biology of each genus, but aims to investigate their potential roles in regulating the abundance of the two size fractions of plankton. (see P18, Lines 20–424 in the revised ms)

**Specific comments:**

Abstract.

P1 – 10. The word 'trophism' is introduced yet does not technically mean what the authors define it as (food resources and trophic levels), and is not used within the field's jargon as such either. I would prefer 'trophic structure' or 'trophic interactions', or 'trophic preferences'.

Response: Agree to change. We have changed into ″trophic preference″ in the revised version. (see P1, Line 11 and P4, Line 1 in the revised ms)

'Temperature-related' seasonal variations – The effect of temperature from season was not separated in this study, it should be simply 'seasonal variations'.

Response: Agree to change. (see P1, Lines 14–15 in the revised ms)

Introduction

P2-0. "With broad feeding spectra and flexible feeding strategies, the bulk copepod assemblage is omnivorous depending on dominant species or group". Omnivo- rous or what? Consider changing to something like 'displays varying degrees of herbivory/ omnivory/carnivory, depending on dominant: : :'

Response: Agree to change. (see P2, Lines 4–5 in the revised ms)

P2-5. "In turn, TLs of a diverse: : :". I assume the authors here refer to the average trophic position of the assemblage, and thus should be 'TL' (singular). "Because copepods play a fundamental role in feeding on phytoplankton as primary consumers". Consider re-phrasing as 'Because copepods rely significantly on phytoplankton as prey', otherwise the expectation of this phrase is that the second half will refer to the top down effect of copepods on phytoplankton, and not the bottom-up effect of phytoplankton on copepods. 'feeding on phytoplankton as primary consumers, so the seasonal and spatial'. Delete 'so'.

Response: Agree to change. (see P2, Line 9 in the revised ms)

P2-15. "Therefore, the assessment of the trophic position (: : :) of copepods within a complex planktonic food web is critical in predicting the ecological relationships between predator and prey". This phrase seems redundant, isn't the study about assessing these ecological relationships? I don't understand the prediction part.

Response: To avoid confuse, we have revised ″in predicting″ to ″to understand″. (see P2, Line 21 in the revised ms)

P3. 0. "In contrast, the d15N values of primary producers increase from being nutrient sufficient (high fractionation) to nutrient-limiting (low fractionation) and are especially high in anthropogenic wastewater nitrogen inputs". Would the later simple swamp the

fractionation effect? The literature on ïA¸d'15N of different nutrients in the ocean (nitrate, ammonia, urea) shows ranges that are much larger than fractionation factors, e.g. these vary by about 20‰ compared to 3.4‰ of fractionation. Can you comment on how much you expect the source to vary along the river gradient?

Response: The parentheses here indicate the consequences. For example, rich nutrients will cause high fractionation of primary producer. And they would continuously accumulate their $\delta^{15}N$ in the cells. We don't have the data on the variations of $\delta^{15}N$ of source, as we expect the$\delta$15N of source will vary from 0 to 13‰ based on the variances of POM. (see P3, Line 5–7)

Materials and methods

P4-25. Could you mention the average volume filtered per tow, as the net was equipped with a flow meter?

Response: Sure, we can provide this. (see P5, Lines 3–4 in the revised ms)

P5-5. "water samples were transported to the laboratory as soon as possible". Please give a time estimate.

Response: 1–2 hours driving. (see P5, Line 14 in the revised ms)

P6-5. The analytical precision of 0.2‰ and 0.3‰ for d13C and d15N, respectively, seems a bit high. Could you estimate what is the lowest change in TL that you can estimate based on this instrument error?

Response: We have re-checked the precision of the instrument during the period we measured the samples. The analytical precision should be 0.1 and 0.05 ‰ for $\delta^{13}C$ and $\delta^{15}N$, respectively. Based on the equation of trophic level, the lowest change will be less than 0.02 TL. (see P6, Line 11 in the revised ms)

P6.15. The weight difference between cyclopoids and calanoids was generated randomly. I don't understand why the information from the species identification was

not used for this purpose. What is the error associated with this type of computation? I would really suggest the authors do a literature search of the mean weights of the difference species and genera enumerated in their samples, and use this information to estimate both cyclopoid/calanoid weights, and the weights of the different calanoid genera. If the composition has already been estimated, it makes no sense to make these assumptions that only introduce greater error into an already indirect way of estimating species stable isotope composition.

Response: As mentioned above, we have doe a literature search. (see Supplementary Table S1 and P6, Lines 13–22 in the revised ms)

P7-20. 'fractionation factors used in the model estimation were calculated from TLs'. I don't understand this statement, it sounds like 3‰ and 0.5 ‰ were assumed (logically) and not calculated. Please clarify.

Response: The sentence of "fractionation factors used in the model estimation were calculated from TLs' means that the fractionation factor between two trophic levels, based on the difference of each $\delta^{15}N$ value. Or it can also be understood by this way, it is multiplying the difference of two trophic levels (calculated from different $\delta^{15}N$ value) with 3‰ and 0.5 ‰ for $\delta^{15}N$ and $\delta^{13}C$, respectively. (see P7, Lines 22–23 in the revised ms)

Results

The authors discuss their seasonal results in the context of 'temperature'. I would prefer to see this discussed as 'seasonal', since temperature variability within a season was not tested and hence the driver of the observed effects cannot be unequivocally stated to be temperature. Rather, they are probably a combined effect of the changes that co-occur with each season and should be stated as such.

Response: We do test the partial effect of temperature on variation of plankton isotope based on GAM results (Fig.3). However, we agree to avoid too much emphasis on temperature but on season. (see Supplementary Fig. S1 and P9, Linse 13–32 in the

revised ms)

P8-10. 'Despite insignificant spatial variability, higher Chl a concentrations generally occurred in the middle of the bay'. This is not obvious from the values in the table. Please explain in more detail or remove.

Response: Agree to revise. (see P8, Line 8–12 in the revised ms)

P8-25. Please give a mean value for copepod d13C as done for the groups above (nanoplankton and microplankton)

Response: Agree to revise. (see P8, Line 30 in the revised ms)

P9-0. "Overall, seasonal succession of winter-spring, spring-summer, and summer autumn were apparent for all plankton groups". Not clear what this means. There appears to be significant overlap in values for the nanoplankton, and no clear increasing progression from winter to autumn, as increases/decreases seem to interchange.

Response: Yes, seasonal pattern for nanoplankton was not clear. We decide to delete this sentence. (see P9, Line 6 in the revised ms)

P9-5. It isn't clear to me how the coefficients of variation are calculated. The range of d15N values encountered is less than that for d13C, although the spatial progressions are less monotonic. Please clarify in the methods how this is calculated.

Response: We calculated the coefficients of variation by dividing the standard error with mean value. We have tried to clarify this in the M&M. (see P8, Line 2 and P9, Line 7 in the revised ms)

P9-10. The result for the microplankton is inconsistent with the figure. In the figure, the highest value for d15N is 10‰ at the bay in spring. There is no 16.2 value.

Response: We may have a typo. It should be 10.2 at station 9. Thanks for the reviewer's careful check. (see P9, Line 18 in the revised ms)

P9-15. "Copepod d15N : : : being much more consistent with the pattern of microplankton than that of nanoplankton". This seems true for the summer ïA̧d'15N values, and quite the opposite for the winter values. Regardless, there is such high variability that it is hard to tease out any clear pattern of spatial/seasonal co-variability.

Response: Yes, it is true. We have removed this unsuitable sentence. (see P9, Lines 25–27 in the revised ms)

P9-20. The GAM result is very interesting. Perhaps it reflects the food-web processes that affect d15N disproportionally and were not included in the GAM?

Response: Here the deviances explained by GAM suggest those factors combined to influence the dependent factor. For $\delta^{15}N$, the deviance explained was relatively lower suggesting that other factors which were not included would contribute another 23% of the deviance of $\delta^{15}N$. But we don't know what are them. The understanding of the reviewer is right. (see Table 2, Table 3, and P9, Lines 25–P10, Line 32 in the revised ms)

P10-20. It is not clear to me how the trophic levels of brackish copepods can be calculated, when their 13C values do not support the sampled nanoplankton and/or microplankton as their food source. I also don't understand how later in figure 6 they show up enriched, but in figure 4 they are depleted with respect to this food source. The differences between these two figures should be stated clearly as they show different results.

Response: The brackish copepods in this study were defined by empirical taxonomy, including *Pseudodiaptmus* and *Sinocalanus*. The trophic levels were calculated by the formula shown in M&M and figure legend. We admit that there may be some confuse, while we don't think the two figures show different results. Firstly, the results of brackish copepods were averaged from *Pseudodiaptmus* and *Sinocalanus*, while our result showed more insight information that they were different.

Secondly, as mentioned above, we didn't intend to give detail information of the biology of each genus, but aims to investigate their potential roles in regulating the abundance of the two size fractions of plankton. Thus, we were unable to provide detail information of all potential diet sources of them, whereas they still have enrichment factor on lower trophic levels such as nanoplankton and microplanton. (see P10, Lines 25–32 in the revised ms)

P10-25. "The enrichment values for nanoplankton feeding on marine and brackish water calanoids: : :". This phrase says that nanoplankton are feeding on copepods. That's not right, it should say something like 'enrichment values for marine and brackish water: : : feeding on nanoplankton'.

Response: Sorry for a mistake here. We have revised as suggested. (see P11, Lines 6–12 in the revised ms)

P11-5. I disagree with the statement (based on the figure) that "the proportions of the two size fractions of POM averaged from all four seasons contributing to copepod diets at different stations were also distinctly different except for station 8 (Fig. 8)". It seems that the error bars overlap at station 1 (hence not different), and stations 6 and 7. I might be missing something but then it should be clarified.

Response: Yes, we found that error bars were indeed overlapping. We have revised this conclusion. (see P11, Lines 19–30 in the revised ms)

P11-10. Does 'spring data available' mean 'only spring shown'?

Response: Yes, we obtained enough amounts of decapods for isotopic analysis only at the spring. However, as suggested by another reviewer, we decide to delete the part of decapods as it was not related to the topic of this study. (see P11, Line 27 in the revised ms)

The authors discuss size-selective feeding of calanoids in the context of 'filtering

efficiency', yet they are not true filter feeders, they are suspension feeders that trap and handle particles (Paffenhofer et al, 1982, Mar Bio 67:2), which has different implications for particle handling. This is an important distinction that should be observed throughout the MS.

Response: OK, we have carefully checked the whole MS and change to "feeding efficiency" (see P17, Line 7 in the revised ms)

Discussion

P13-0. It seems to me that the sewage explanation deserves a bit more attention. If the authors can't rule it out it means that this could contribute substantially and swamp the other subtle processes discussed in the 15N-enriched ammonia section.

Response: Yes, we also believe that sewage was important for $^{15}$N accumulating. However, we didn't have direct data to support our speculation. Thus we have changed the sentence to "The input of sewage-derived $^{15}$N-enriched ammonia 5 (domestic sewage and livestock waste) could contribute substantially and swamp the other subtle processes to increase $\delta^{15}$N values of nanoplankton". (see P13, Line 20–22 in the revised ms)

P13-5. "Furthermore, the fractionation effect of phytoplankton will be reduced when phytoplankton became nitrogen-limited and take up nitrogen with little fractionation". I am unsure that this effect could be significant in a coastal areas such as this one. Moreover, if phytoplankton reduce their fractionation, it would mean that their $^{15}$N will tend to be higher (as they choose the lighter $^{14}$N), and thus doesn't explain this decreasing trend.

Response: Yes, we agree that nutrient-limiting is not frequently happened in coastal area. However, substantial reduction of nutrients from different seasons or from different stations and the mis-match of high phytoplankton and low nutrients were normal. When phytoplankton reduce fractionation, they will select more ligther $^{14}$N in cells thus they will show a reducing ratio of $^{15}$N in cells (Cifuentes et al., 1988; Fogel

and Cifuentes, 1993; Granger et al., 2004). To remove such confuse, we have revised this sentence to "Furthermore, the fractionation effect of phytoplankton will be reduced when nutrients substantially decreased and phytoplankton would take up nitrogen with little fractionation and stored relatively light of nitrogen isotope." (see P13, Lines 24–25 in the revised ms)

P13-10. I would like to see table with the GAM results. It would be nice to have these presented first in the results, and later discussed. It would also be interesting to see the different variables tested and the ones found to be significant within this table.

Response: Agree to revise. We have tried to show the table in Results. And as suggested by another reviewer, we will move the GAM figures (Fig. 3). (see P9, Lines 13–32, P13, Lines 12–P14, Line 12, and Tables 2–3 in the revised ms)

P13-20. But see Gutierrez-Rodriguez et al (2014, L&O, vol:59, i5) on negligible trophic enrichment of heterotrophic protists.

Response: Thank the reviewer's reference. We agree the negligible trophic enrichment of heterotrophic protists, thus we have removed such speculation in discussion of the revised ms.

P14-0. "Because of different feeding behaviors and fractionation effects of copepods, the variability of trophic positions of copepod assemblage depends on the overall composition of species and is determined by dominant species." Change to ": : :the variability of the average community trophic position depends on the overall composition of species and is determined by the dominant species."

Response: Agree to revise. (see P14, Lines 13–14 in the revised ms)

I am somewhat confused about the discussion of trophic levels of the copepods Paracalanus and Sinocalanus. The authors state that their ïAˌd'13C values are lower than all measured food sources, which would imply that their food source has not been

adequately measured. How then are these organisms included in the trophic level (TL) component of the paper? A bit of clarification on this topic would really help the reader.

Response: Agree to do so. The trophic level in this study was defined as trophic position relative to nanoplankton, which was considered as the trophic baseline. (see P10, Lines 25–26 in the revised ms)

P17-10. This paragraph explaining the Bayesian mixing model methods/results should be moved to the results section.

Response: Agree to revise. Instead, we have added some more discussion on this part. (see P17, Lines 26–P18, Lines 4)

**Reply to Referee #4**

**General Comments**

This manuscript provides results from seasonal and spatial variation in the stable isotopes 13C and 15N of POM and copepods along a salinity gradient in Gwangyang Bay, off the southern coast of Korea. The authors combined this information with linear mixing models, Bayesian isotopic mixing models and generalized additive models to derive a statement on food selectivity and trophic level of copepods. In general, this manuscript is very well structured and provides valuable information on the flow of matter through the food web. Still, some concerns have to be clarified before publication.

Response: We appreciated the positive comments of the reviewer and have followed the suggestion to improve the manuscript.

**Specific comments.**

Introduction

1. Page 3, line 7: Please give more information here on the usage of different N sources and enrichment factors.

Response: Agree. Accordingly, we have explained more information here based on literature. (see P3, Lines 8–11 in the revised ms)

2. Page 3, line 19: "highly mixed species"- Please clarify, mixed with what?

Response: Here the "highly mixed species" means the assemblage contained too many different species and those species had similar size. So such species were hard to be sorted out from the assemblage based on current microscopic technique. To remove confuse, we have revised it to "high diversity of the assemblage and …". (see P3, Line 21 in the revised ms)

3. Page 3, line 21: Instrument sensitivity has increased and compound specific analysis (CSI) of stable isotopes in amino acids make it possible to track diets of mesozooplankton and determine their trophic position.

Response: Yes, of course. We admit that highly developed instrument can do so. But for doing so, researchers still need taxonomic expertise to sort out the species from a complex mixture to prepare the sub-sample. It requires a lot of lab processing works. (see P3, Lines 24–28 in the revised ms)

4. Page 3, line 21: Please give some reason why this site was chosen.

Response: The stations were chosen based on salinity regime and different geographic characteristics, e.g. stations 1–3 are river sites with extremely low salinity, stations 4–6 are in the central bay with moderate salinity, while stations 7–9 are in the channel towards to the open ocean with relatively high salinity (see P4, Lines 26–29 in the revised ms).

Material and Methods

5. General: why did the authors not use literature data on average weight values for each of the species investigated instead of assigning the weight to each group?

Response: In the revised version, we have searched for the literature data just like suggestion of this comment and also suggested by other reviewers. (see Supplementary Table S1 and P6, Line 13–22 in the revised ms)

6. General: How where copepodite stages treated regarding abundance and body mass?

Response: They were averaged to adults. (see P5, Line 9 in the revised ms)

7. Page 4, line 15: Change to "increasing".

Response: Agree and we have revised accordingly. (see P4, Line 16 in the revised ms)

8. Page 4, line 16: Specify "in the middle of Gwangyang Bay.

Response: Agree. We have revised to "in the middle part of the Gwangyang Bay". (See
        P4, Line 18 in the revised ms)

9. Page 4: Please add information on when sampling took place- day or night?

Response: We all sample at the day time. We have add edsuch explanation in M&M. (see
        P5, Line 1 in the revised ms)

10. Page 5, line 11: "pico- and nano- sized phytoplankton". Doesn' t sampling with a
mesh also include nanozooplankton like heterotorphic and mixotrophic flagellates- so
it does not only comprise phytoplankton?!

Response: Here the plankton less than 20 micron but larger than GF/F (0.78 micron) were
        defined as nanoplankton. Thus they contains both phytoplankton and heterotrophs.
        (see P5, Lines 13–17 in the revised ms)

11. Page 6, line 29: something is missing at the end of the sentence- "illustrated in
figures?".

Response: The figures here do not mean citations. We try to explain that the mean and
        standard deviations were illustrated by forms of figures. To remove confuse, we can
        delete this sentence in the revised version. (see P6, Line 30 in the revised ms)

Results and Discussion

12. There are too many figures. Some might be moved to the supplemental section, e.g. Fig. 3, 7,8,11

Response: We agree to do so. We have moved Figs. 3, 7, and 8 to supplementary materials, but no Fig.11. We believe that Fig.11 is relatively important for readers and other reviewer want to know more about the information of the feeding of carnivorous species. (See Supplementary Figs. S1–S3 and Fig. 8 in the revised ms)

13. Page 12, line 16: What is a "heavy carbon pool", give an example?

Response: The phrase is located at "Page 12, line 24". "Heavy carbon pool" here means the dissolved inorganic carbon pool in which the carbon was primarily composed by heavy carbon ($^{13}C$). (see P13, Lines 5–7 in the revised ms)

14. Page 12, line 31: Wording! Please revise "much reduced".

Response: We have changed it to "low". (see P13, Line 16 in the revised ms)

15. Page 13, line 15: "with low fractionation effects"- give example.

Response: Now we have deleted this kind of discussion about temperature in the revised ms, as suggested by other reviewers.

Conclusion

16. Please provide a simplified figure of the energy flow for the different seasons.

Response: Based on revised estimation, we have tried to provide such simplified figures. (see Fig. 9 and P8, Line 20–24in the revised ms)

I hope that these revisions are satisfactory and that the revised version will be acceptable for publication in Biogeosciences.

Sincerely yours,

Chang-Keun Kang

[revised manuscript text omitted]